# 🔪CLIPPER: Compression enables long-context synthetic data generation

**Chau Minh Pham[1]    Yapei Chang[1]    Mohit Iyyer[1,2]**

[1]University of Maryland, College Park    [2]University of Massachusetts Amherst

{chau,yapeic,miyyer}@umd.edu

## Abstract

LLM developers are increasingly reliant on synthetic data, but generating high-quality data for complex long-context reasoning tasks remains challenging. We introduce CLIPPER,[1] a compression-based approach for generating synthetic data tailored to *narrative claim verification* – a task that requires reasoning over a book to verify a given claim. Instead of generating claims directly from the raw text of the book, which results in artifact-riddled claims, CLIPPER first compresses the book into chapter outlines and book summaries and then uses these intermediate representations to generate complex claims and corresponding chain-of-thoughts. Compared to naïve approaches, CLIPPER produces claims that are more valid, grounded, and complex. Using CLIPPER, we synthesize a dataset of 19K claims paired with source books and chain-of-thought reasoning, and use it to fine-tune three open-weight models. Our best model achieves breakthrough results on narrative claim verification (from 28% to 76% accuracy on our test set) and sets a new state-of-the-art for sub-10B models on the NoCha leaderboard. Further analysis shows that our models generate more detailed and grounded chain-of-thought reasoning while also improving performance on other narrative understanding tasks (e.g., NarrativeQA).

○ https://github.com/chtmp223/CLIPPER

## 1 Introduction

Due to the high cost of human-annotated data, LLM developers increasingly rely on *synthetic* data (generated by LLMs) to boost instruction following and reasoning capabilities (Ding et al. 2023; Lambert et al. 2024; Yang et al. 2025, *inter alia*). As the context size of LLMs extends to millions of tokens, it is important to ensure that we have scalable and performant strategies to create synthetic data for *long-context* tasks. Prior work creates such data by (1) selecting a long document or a smaller chunk within; and (2) prompting an LLM to generate input/output pairs using the selected text (Bai et al., 2024; Dubey et al., 2024).

While this strategy is effective for tasks like summarization and QA, we show that it breaks down for more complex reasoning-oriented tasks like *narrative claim verification*, in which a model must judge whether a statement about a long input text is true or false. The majority of narrative claims in the NoCha benchmark (Karpinska et al., 2024), which was created by human readers of fictional books, can only be verified by *global* reasoning over events, characters, and relationships.[2] This poses a challenge to even the best LLMs: OpenAI's o1-preview currently leads with an accuracy of 67.4% (far below human performance). If no LLM can reliably solve the task, how can we produce and validate synthetic data for it?

We tackle this challenge by introducing CLIPPER, a synthetic data generation pipeline that operates in two stages (Figure 1). First, a long document is *compressed* by an LLM into

---

[1]CLIPPER stands for **C**ompressing **L**ong **I**n**P**uts.
[2]Creating NoCha is costly and challenging: annotators read full books and earn $1.70 per claim.

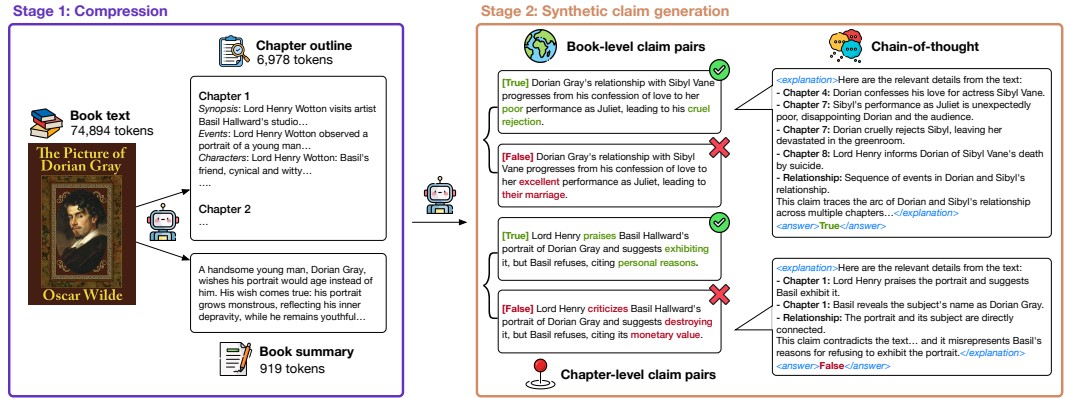

Figure 1: CLIPPER overview. (1) *Compression*: An LLM generates chapter outlines (8.7K tokens) and summaries (618 tokens) from books (90K tokens). (2) *Claim generation*: The LLM produces true/false claims with chains-of-thought with the outlines and summaries.

summaries and/or outlines that contain salient events. Then, the LLM generates claims based on the compressed narrative, with optional instructions to write claims that require reasoning over multiple chapters to verify. Each claim is accompanied by a chain-of-thought reasoning trace that grounds it within specific chapters where relevant events occur.

Compared to the naïve strategy of prompting an LLM with the entire (uncompressed) book, CLIPPER significantly reduces the error rate in the claims from 73.1% to 16.7%, while also producing more claims at a lower cost. Why does this work? Prior work has shown that LLMs are high-quality summarizers of long documents (Chang et al., 2024; Kim et al., 2024). By operating on compressed representations, we also address the known degradation of instruction-following in long-context settings (Wu et al., 2024b; Levy et al., 2024) and thus reduce the complexity of the claim generation process.

We use CLIPPER to generate a dataset containing 19K claims about public-domain fictional books. Fine-tuning open-weight models like Llama-3.1-8B-Instruct (Dubey et al., 2024), ProLong-512K-8B-Base (Gao et al., 2024c) and Qwen2.5-7B-Instruct (Qwen et al., 2024) on this dataset yields large improvements on narrative claim verification and positive transfer to other narrative-related tasks (e.g., NarrativeQA, MuSR). For instance, fine-tuning Llama-3.1-8B-Instruct on our data doubles its NoCha performance (from 16.5% to 32.2%) and almost triples its test set performance (from 27.9% to 76.0%). Our fine-tuned Qwen model sets a new state of the art on NoCha for <10B models, outperforming closed models like Gemini 1.5 Flash 8B (Team et al., 2024) and OpenAI o1-mini (OpenAI, 2024).

While our approach is promising for improving long-context reasoning in open-weight models, our best models fall well short of the performance reached by closed LLMs such as o1 on NoCha. The performance gap between NoCha and CLIPPER-test likely stems from the nature of the claims, as CLIPPER-test features synthetic claims based on model-generated outlines, while NoCha's human-written claims require reasoning about details often absent from such outlines. We analyze where our fine-tuned models can still improve, discovering that they benefit more from training on claims whose evidence is localized to a single chapter (and not more complex multi-chapter claims). We hope future work will use our data generation pipeline for fine-tuning larger models (e.g., >70B) on other long-context tasks to further improve global reasoning abilities. In summary, our contributions are:

1. CLIPPER: A compression-based pipeline for synthesizing grounded long-context data at low cost, producing 19K claim-book pairs for narrative claim verification.
2. Fine-tuned open-weight models that advance claim verification and narrative understanding, showing the benefits of training on data generated with CLIPPER.

## 2 CLIPPER: generating high-quality synthetic data via compression

In long-context settings, synthetic datasets have typically been created by selecting lengthy documents from an existing corpus and using an LLM to generate input-output pairs given either the entire document (Bai et al., 2024) or random excerpts (Dubey et al., 2024; Yang et al., 2025). For our task, however, we show that these methods are insufficient:

- *Providing the LLM with the entire document* results in much noisier data, as we show that producing high-quality, complex claims about long narratives is a fundamentally difficult task even for the best models (§2.2).
- *Providing only an excerpt from the long document*, on the other hand, precludes the model from generating claims that require global reasoning across the entire book.

We thus develop a two-stage strategy, CLIPPER, which first compresses the narrative into chapter outlines and summaries, then prompts an LLM to produce claims and chain-of-thoughts grounded in the compressed narrative (§2.3). We use the dataset generated by CLIPPER to fine-tune open-weight models in §3. **All prompts can be found in §A.5.**

### 2.1 Task setup

Before describing how CLIPPER works, we first establish the definition for *narrative claim verification*, then explain how we collect the books that serve as the foundation for this task.

**Task definition:** In *narrative claim verification*, an LLM is given a book and a claim about the book. The task is to determine whether the claim is true or false while providing a clear explanation for its decision. A key aspect of the task is the inclusion of **true/false** *narrative minimal pairs* (Karpinska et al., 2024), where each false claim closely resembles its true claim counterpart but contains subtle inaccuracies (illustrated in Figure 1; see Table 4 for more examples). The model is considered accurate only if it correctly verifies both claims in a pair, which reduces the chances of the model being correct for the wrong reason.

**Gathering public domain books:** We collect **479** fictional books from Project Gutenberg, with an average length of **90K** tokens[3] and **23** chapters.[4] While these texts might raise memorization concerns, §A.2 shows they do not affect baseline model performance.[5]

### 2.2 Naïve claim generation using book texts

One simple data synthesis approach is prompting an LLM to generate claims directly from the book text. However, this NAÏVE method falls short in a long-context setting, with **73.1%** of the generated claims containing serious errors. In addition, NAÏVE produces fewer claims at a higher price. These limitations motivate us to develop CLIPPER.

**The NAÏVE method:** We provide Claude-3.5-Sonnet-v1 with the entire book text and prompt it to generate pairs of true/false claims along with corresponding chain-of-thought reasoning in a zero-shot manner. Note that we cannot use few-shot prompting due to the books' length. We finally prompt Claude to remove any duplicated claims among the generated claims. Note that we do not ask Claude to validate the generated claims, as this is a much more challenging task (as seen by Claude's 40.3% accuracy on NoCha)—the very problem we aim to address in this paper. In contrast, claim validation is very feasible with CLIPPER, as Claude's claim verification accuracy given outlines instead of text is 98.6%.

---

[3]All token counts are computed using o200k_base from https://github.com/openai/tiktoken.

[4]We do not include books longer than 128K tokens as many open-weight models cannot process anything beyond that number. We clean the manually downloaded books by removing supplementary content to prevent models from using these metadata as shortcuts for event retrieval.

[5]https://chatgptiseatingtheworld.com/wp-content/uploads/2025/01/Unredacted-Reply-of-Plaintiffs-1.pdf shows that Llama models might have been trained on LibGen book data.

| Category | Naïve | Clipper | Error Definition | Example |
|---|---|---|---|---|
| Invalid | 11.5% | 9.1% | The claim is incorrect with respect to the book text, or the true/false claim pair is invalid. | Anne rejects three marriage proposals during her time at Redmond College: from Charlie Sloane, Gilbert Blythe, and Roy Gardner, all because she doesn't love them. *(This false claim is not entirely false because Anne really didn't love them or wasn't initially aware of her romantic feelings.)*. |
| Mis-attribution | 28.9% | 4.6% | The claim is valid, but the associated explanation does not cite the correct chapters. | Dr. Sheppard...was the last person known to have seen Roger Ackroyd alive at 8:50 PM on the night of the murder, and he later assisted Hercule Poirot in the investigation while simultaneously concealing Ralph Paton in a nursing home. *(The explanation cites Chapter 1, 4, 16, and 20, but misses Chapter 24, which mentions that Ralph is in a nursing home)* |
| Explicit references | 15.4% | 0.0% | The claim is easier to verify since it includes direct quotes and chapter references, eliminating the need for event retrieval. | Alice's pursuit of the White Rabbit, which begins with her following him down a rabbit hole in Chapter 1, continues throughout her adventure, including an encounter in the King and Queen of Hearts' court in Chapter 11 where the Rabbit acts as a herald. |
| Duplication | 17.3% | 3.0% | The claim describes the same events as another. Although their content is similar, differences in wording may allow both to pass our deduplication process. | "Dorian Gray's cruel rejection of Sibyl Vane after her poor performance as Juliet leads to her suicide, which Dorian callously dismisses by attending the opera the following night, resulting in the first noticeable change in his portrait [...]" versus "Dorian Gray's cruel rejection of Sibyl Vane after her poor performance as Juliet leads to her suicide, causing the first visible change in his portrait [...] culminate in his murder of Basil Hallward years later [...]." |
| **Any error** | **73.1%** | **16.7%** | The claim is invalid, misattributed, duplicated, or contains explicit references. | |

Table 1: Error types among claims produced by NAÏVE (52) and CLIPPER (66) based on six books from Table 5. Examples are selected from NAÏVE claims.

**Human validation of NAÏVE claims:** We manually annotate 52 claims generated by NAÏVE based on six books (Table 5).[6] Across these claims, we identify four types of errors. Table 1 shows that 73.1% of NAÏVE claims contain an error. Specifically, there are 11.5% *invalid* claims, often due to mislabeled false claims that are actually valid. There is also a high number of *misattributed* (28.9%) claims that cite the wrong chapters in the produced chain-of-thought. 17.3% of the claims are *duplicated* despite the deduplication step, because Claude frequently hallucinates source chapters. 15.4% of the claims also include *explicit references* to chapter numbers or direct quotes,[7] which compromises the task by revealing the evidence location. Additionally, we observe that the events referenced in the generated claims are often the book's most major events, making the claims much easier for LLMs to verify. Beyond these errors, the NAÏVE pipeline is costly at ≈$0.07 USD per claim (≈ $1,330 for 19K claims).

## 2.3 Claim generation with CLIPPER

To produce more valid and grounded claims, we use *compressed* representations of the book content, namely chapter outlines and book summaries. These intermediate forms help anchor claims to specific events in the book, reducing the need to search through the entire text for relevant details. Additionally, this approach makes it easier to generate claims about lower-level events, addressing a major limitation of the NAÏVE approach. We first (1) compress the books into a chapter outline and book summary and then (2) generate pairs of true/false claims at different scopes based on these compressed representations.

**(1) Compressing books into summaries and chapter outlines:** Book summaries provide a global context for claim generation to ensure that each claim is consistent with the entire book. We prompt GPT-4o to summarize the entire book into a few paragraphs (≈ 618 tokens on average). Chapter outlines provide a list of fine-grained events that can be used to construct grounded claims. We prompt Claude[8] with each chapter text to generate an outline containing a synopsis, major events (5–7 per chapter), and a character list. Our

---

[6]We annotate each claim with its most major error type.

[7]This happens despite explicit formatting instructions.

[8]We set temperature=0.0, max_tokens=4096. We use Claude instead of GPT-4o because Claude includes more concrete and objective events for the outline.

compression rate is 10.0%, calculated by averaging the ratio of outline length (8,745 tokens on average) to full book length (90,437 tokens) across all books.

**(2a) Generating claims from compressed narratives:** We use chapter outlines and book summaries to generate true/false claims. We synthesize claims at two different scopes to enable reasoning across different token ranges:

> **Book-level claims:** Claude is prompted to identify 2–3 key events from the outlines of at least 2 chapters, then use them to generate a claim. These claims require models to have a global understanding spanning multiple parts of the book.

> **Chapter-level claims:** Given the book summary and a single chapter outline, Claude is instructed to identify 2-3 key events of the chapter and use them to write a claim. While these claims do not necessitate global reasoning, they still require the model to search for correct chapter within a long text and perform intra-chapter reasoning (§4.5).

**(2b) Deduplicating and validating generated claims:** Just as in NAÏVE, we use Claude to remove duplicate claims. Additionally, we use GPT-4o[9] to validate the claims against the source chapter outlines by prompting it to assess whether all parts of a claim are supported by the outline. This step is another advantage of CLIPPER: unlike the NAÏVE approach, our method allows for claim verification using the compressed chapter outline. To evaluate the reliability of LLM-based filtering, we manually review 72 claim pairs and only disagree in one instance, where GPT-4o deems a claim valid that we find too subjective. Overall, 59.4% of the original claims are removed as duplicates, while 2.4% are filtered out as invalid.

**Human validation of CLIPPER claims:** We use the same setup as described in §2.2 to manually evaluate 66 claims generated by the CLIPPER pipeline. Table 1 provides a detailed breakdown of issues flagged in these claims, such as explicit references, invalidity, duplication, or misattribution. Notably, **83.3% of the 66 claims are found to be completely free of errors—a significant improvement compared NAÏVE's 26.9%**. CLIPPER also costs less at $0.05 USD per claim compared to NAÏVE ($0.07 per claim) and human annotators ($1.7 per claim based on Karpinska et al. 2024).[10]

## 2.4 Automatic validation of CLIPPER chain-of-thoughts

We assess the groundedness of chain-of-thought (CoT) reasoning by prompting an LLM to verify whether each event in the CoT is supported by the chapter outline. Accuracy is measured as the percentage of events in true claim CoTs that are grounded in the book. To scale up evaluation, we use an LLM judge, DeepSeek-R1-Distill-Llama-70B (DeepSeek-AI et al., 2025). We find that 98.5% of CoTs are grounded. The remaining ungrounded CoTs typically involve events open to multiple interpretations. Compared to NAÏVE, CLIPPER's CoTs are significantly easier to verify due to their explicit chapter references.

## 3 Supervised fine-tuning for LLMs on CLIPPER data

Having shown that CLIPPER produces synthetic data of high quality, we now investigate the effects of training on such data. We apply supervised fine-tuning (SFT) to three models on our dataset: ProLong-512K-8B-Base (Gao et al., 2024b),[11] Llama-3.1-8B-Instruct (Dubey et al., 2024), and Qwen2.5-7B-Instruct (Team, 2024).[12] Our top model, Llama-CLIPPER, achieves nearly three times the test set performance of Llama-Instruct—boosting accuracy from 27.9% to 76%—while showing substantial gains in long-context reasoning and narrative

---

[9]Chosen to mitigate potential self-biases (Xu et al., 2024b; Panickssery et al., 2024; Li et al., 2025).

[10]See detailed cost analysis in §A.4.

[11]Despite the name, this model has undergone instruction tuning before. The ProLong team ran continual pre-training on Llama-3-8B-Instruct to get this model.

[12]We will now refer to these models as ProLong-Base, Llama-Instruct, and Qwen-Instruct.

understanding on tasks like NoCha, NarrativeQA, and MuSR. Moreover, all of our models outperform all existing <10B models on the NoCha benchmark.

## 3.1 Training setup

**Data splits and hyperparameters:**   We divide our dataset into three parts: 16K claims (8K true/false pairs) for training, 2K for validation, and 1K for testing. Notably, the books in the test set do not overlap with those in the training or validation sets. For each entry, we combine the book text and claim to form the user prompt, and include the chain of thought reasoning along with the final answer as the assistant's message (see Figure 16). A learning rate of 1e-6 and a batch size of 16 yield the best performance on our dev set.[13] We fine-tune Qwen-Instruct, Llama-Instruct, and ProLong-Base using this configuration for one epoch.

**Ablation on the effect of claim scope:**   Our dataset consists of 8K book-level and 8K chapter-level claims. We fine-tune ProLong-Base separately on each claim scope subset, resulting in ProLong-`CLIPPER`-chapter and ProLong-`CLIPPER`-book.

**Ablation on the effect of data length:**   Prior work shows that fine-tuning on short texts can improve long-context performance in tasks like QA and summarization (Dubey et al., 2024; Gao et al., 2024b). Since our dataset contains long documents averaging 90K tokens, we test whether short-text fine-tuning also helps with long-context claim verification. We use WritingPrompts (Fan et al., 2018), a dataset of 300K stories averaging 742 tokens, and extract claims directly without generating outlines or summaries.[14] We collect 19K claims and train on ProLong-Base to get ProLong-WritingPrompts.[15]

## 3.2 Evaluation

Beyond claim verification, we expect that training on our synthetic dataset will also improve performance on related tasks. Therefore, we include both reasoning and narrative understanding benchmarks that vary in input lengths and tasks.

**Claim verification:**   To measure accuracy, we calculate the percentage of cases in which a model correctly verifies both true and false claims within a given pair.

➤ **`CLIPPER-test`** contains 1,000 true/false claim pairs drawn from 53 books, evenly split between book-level and chapter-level claims.

➤ **NoCha** (Karpinska et al., 2024) consists of 1,001 true/false claim pairs about recent fiction books (up to 336k tokens). These claims, crafted by annotators familiar with the books, are much harder to verify compared to those in `CLIPPER`-test.

**General narrative understanding:**   We use three existing benchmarks as detailed below.

➤ **NarrativeQA** (Kočiský et al., 2018) is a long-form Q&A benchmark that requires models to process entire books or movie scripts to answer provided questions. The benchmark consists of 1,572 stories and summaries as well as 46,675 human-written questions. We use the HELMET implementation (Yen et al., 2024) for this benchmark.

➤ **∞Bench QA** (Zhang et al., 2024) is a long-form Q&A benchmark that requires models to answer 351 questions about novels. We use the HELMET implementation but use GPT-4o's judgment as a metric instead of ROUGE F1 (see §C.1 for explanation).

---

[13]We perform hyperparameter tuning on learning rates of 1e-5, 1e-6, and 1e-7, along with batch sizes of 16 and 32. Tuning is done for one epoch on a subset of 2K training samples. Due to high GPU costs (each epoch takes 5 hours), we only conduct hyperparameter tuning on ProLong-Base only.

[14]We use a prompt similar to the one in §2.3.

[15]After doing hyperparameter tuning on 2K training samples, we decide on the learning rate of 1e-5 and batch size of 16 as the best training configurations. We tested learning rates of 1e-5, 1e-6, 1e-7 and batch sizes of 8, 16, 32, 64.

| Models | CLIPPER-test | NoCha | NarrativeQA | MuSR | ∞Bench QA |
|---|---|---|---|---|---|
| Qwen2.5-7B-Instruct | 51.0% | 24.1% | 40.3% | 41.2% | 35.3% |
| Llama-3.1-8B-Instruct | 27.9% | 16.5% | 47.7% | 40.3% | **47.8%** |
| ProLong-512K-8B-Instruct | 34.5% | 16.9% | 44.0% | 42.3% | 42.6% |
| 🔊 Qwen2.5-7B-CLIPPER | 73.9% | **32.4%** | 46.0% | **45.2%** | 42.3% |
| 🔊 Llama-3.1-8B-CLIPPER | **76.0%** | 32.2% | **49.0%** | 43.6% | 46.5% |
| 🔊 ProLong-512K-8B-CLIPPER | 75.0% | 32.3% | **49.0%** | 44.5% | 38.5% |
| ProLong-512K-8B-WritingPrompts | 63.0% | 24.1% | 31.0% | **45.2%** | 35.8% |

Table 2: Model accuracy on claim verification (CLIPPER-test, NoCha) and narrative understanding benchmarks (NarrativeQA, MuSR, ∞Bench QA). Fine-tuning models using CLIPPER improves performance on claim verification and narrative understanding.

➤ **MuSR** (Sprague et al., 2024) includes 756 algorithmically generated problems such as murder mysteries, object placement questions, and team allocation optimization. We use the LM Harness (Gao et al., 2024a) implementation.

# 4 Results & analysis

Our fine-tuned models set a new state of the art for <10B models on long-context claim verification while also improving baseline performance on narrative understanding tasks.

## 4.1 CLIPPER models outperform baselines on narrative claim verification

On CLIPPER-test, our fine-tuned models significantly outperform the instruct models they are initialized from (referred to as baselines),[16] as shown in Table 2.

For example, Qwen-CLIPPER achieves over a 20% performance gain compared to Qwen-Instruct, while Llama-CLIPPER sees nearly triple the performance of Llama-Instruct. These substantial improvements demonstrate the effectiveness of CLIPPER-generated data.

**Fine-tuning on our data improves performance on NoCha:** A similar trend is observed on NoCha. The performance improvements range from an 8% gain for strong baselines like Qwen-Instruct to a twofold increase for weaker baselines such as Llama-Instruct and ProLong-Instruct. It is worth noting that all baseline models initially perform below the random chance baseline of 25%, but our fine-tuned models consistently surpass this threshold.

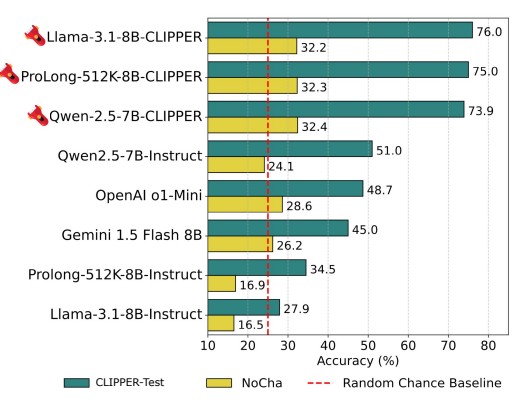

Figure 2: Results on CLIPPER's test set and NoCha for baselines, small closed models, and CLIPPER models. Fine-tuning on our synthetic data significantly improves narrative claim verification.

**Performance gap between CLIPPER-test and NoCha:** We note that the performance gap between NoCha and CLIPPER-test is likely due to the nature of the events involved in the claims. While CLIPPER-test consists of synthetic claims derived from events in model-generated outlines, NoCha's human-written claims may involve reasoning about low-level details that may not typically appear in such

---

[16]ProLong-CLIPPER is initialized from ProLong-Base instead of ProLong-Instruct. However, since performing evaluation intended for instruct models on a continually pretrained model may not be ideal, we exclude ProLong-Base's results from Table 2. As shown in Table 19, ProLong-Base performs significantly worse than ProLong-Instruct on CLIPPER-test.

---

**Claim:** "Jacques consistently uses pastoral themes to express his love for Belle-bouche, first through metaphorical discussions of shepherds and shepherdesses, and later while walking through an orchard and discussing Arcadian lovers."

**Llama-Instruct's CoT:** "The statement claims that Jacques consistently uses pastoral themes to express his love for Belle-bouche. To evaluate this statement, we need to examine the text and identify instances where Jacques uses pastoral themes to express his love for Belle-bouche. The text mentions that Jacques discusses crooks, flowers, and shepherdesses with Belle-bouche, and later, while walking through an orchard, he talks about Arcadian lovers, specifically mentioning Strephon, Chloe, Corydon, Daphnis, and Narcissus. These instances suggest that Jacques does use pastoral themes to express his love for Belle-bouche. However, we need to consider whether this is a consistent pattern throughout the text."
*Answer: TRUE*

**Llama-CLIPPER's CoT:** "Here are the relevant details from the text:
- Chapter 2: Jacques engages in a metaphorical discussion about shepherds and shepherdesses, expressing his love for Belle-bouche.
- Chapter 17: Jacques and Belle-bouche walk through the orchard, discussing the twentieth of May as a day for lovers.
- Relationship: Recurring theme of pastoral love
In Chapter 2, Jacques engages in a metaphorical discussion about shepherds and shepherdesses to express his love for Belle-bouche. This theme continues in Chapter 17, where Jacques and Belle-bouche walk through an orchard, discussing the twentieth of May as a day for lovers. This connection shows Jacques' consistent use of pastoral imagery to convey his romantic feelings for Belle-bouche throughout the story."
*Answer: TRUE*

---

Table 3: Comparison of CoT responses from Llama-Instruct and Llama-CLIPPER for a claim in John Esten Cooke's The Youth of Jefferson. Llama-CLIPPER includes more explicit chapter references (green) and details than Llama-Instruct (red).

generated outlines. Future work could incorporate more low-level events into chapter outlines to create more diverse claims.

## 4.2 Fine-tuning on CLIPPER improves on other narrative reasoning tasks

Beyond long-context reasoning, our models also show improvements in narrative understanding and short-context reasoning tasks. On NarrativeQA, which requires comprehension of movie scripts or full books, our best-performing models, Llama-CLIPPER and ProLong-CLIPPER, achieve a 2% and 5% absolute improvement over their respective baselines. Similarly, on MuSR, a short-form reasoning benchmark, our strongest model, Qwen-CLIPPER, achieves 45.2% accuracy, surpassing the 41.2% baseline. However, on ∞Bench QA, only Qwen-CLIPPER outperforms the baseline by approximately 7%. In contrast, Llama-CLIPPER and ProLong-CLIPPER show slight performance declines of up to 4%. Thus, while fine-tuning on CLIPPER data improves performance on reasoning and some aspects of narrative understanding, its transferability is not universal across domains.

## 4.3 Long-context claim data is more helpful than short-context data

Our results stand in contrast to prior studies suggesting short-form data benefits long-context tasks (Dubey et al., 2024; Gao et al., 2024b) more than long data. While ProLong-WritingPrompts, trained on short data, outperforms baselines, it underperforms across all four long-context benchmarks compared to models fine-tuned on our data. This underscores the need for high-quality long-context data generation pipelines like CLIPPER.

## 4.4 Fine-tuning on CoTs results in more informative explanations

We evaluate the groundedness of CoT reasoning generated by our fine-tuned models using DeepSeek-R1-Distill-Llama-70B (§2.4). Here, a reasoning chain is counted as grounded when every plot event in the chain can be found in the chapter outline that it cites. Table 21 shows that fine-tuning significantly improves groundedness across all models, with ProLong-CLIPPER achieving the highest rate (80.6%), followed closely by Llama-CLIPPER (75.9%). Looking closer at the content of the explanations (Table 3), the baseline model (Llama-Instruct) often gives a generic response without citing any evidence, whereas Llama-CLIPPER explicitly references Chapter 9 and specifies the cause-and-effect relationship.

## 4.5 Small models struggle with book-level reasoning

Trained only on 8K chapter-level claims, ProLong-CLIPPER-chapter outperforms ProLong-CLIPPER-book on both chapter- and book-level test subsets (Table 20). This likely reflects the limitations of smaller models (7B/8B) in handling the complex reasoning required for book-level claims, in line with prior findings (Qi et al., 2024). The performance gap between

| CATEGORY | FREQ (%) | TRUE CLAIM | FALSE CLAIM |
|---|---|---|---|
| Event | 43.2 | The Polaris unit, initially assigned to test a new audio transmitter on Tara, explores the planet's surface using a jet boat without landing. | The Polaris unit, initially assigned to test a new audio transmitter on Tara, explores the planet's surface by landing their spaceship. |
| Person | 31.6 | The cattle herd stolen from Yeager by masked rustlers is later found in General Pasquale's possession at Noche Buena. | The cattle herd stolen from Yeager by masked rustlers is later found in Harrison's possession at Noche Buena. |
| Object | 15.8 | The alien structure Ross enters contains both a chamber with a jelly-like bed and a control panel capable of communicating with other alien vessels. | The alien structure Ross enters contains both a chamber with a metal bed and a control panel capable of time travel. |
| Location | 13.7 | Costigan rescues Clio twice: first from Roger on his planetoid, and later from a Nevian city using a stolen space-speedster. | Costigan rescues Clio twice: first from Roger on his planetoid, and later from a Triplanetary city using a stolen space-speedster. |
| Time | 6.3 | Jean Briggerland's meeting with ex-convicts Mr. Hoggins and Mr. Talmot, where she suggests a burglary target, follows a failed attempt on Lydia's life involving a speeding car on the sidewalk. | Jean Briggerland's meeting with ex-convicts Mr. Hoggins and Mr. Talmot, where she suggests a burglary target, precedes a failed attempt on Lydia's life involving a speeding car on the sidewalk. |
| Affect | 4.2 | David Mullins, who initially expresses skepticism about Chester's hiring, later fires Chester on false pretenses and immediately replaces him with Felix. | David Mullins, who initially expresses enthusiasm about Chester's hiring, later fires Chester on false pretenses and immediately replaces him with Felix. |

Table 4: Taxonomy of perturbations causing false claims to be misclassified as true. True and false details are highlighted in green and red, respectively. Frequencies may exceed 100% due to multi-labeling. See §D.2 for definitions and analysis.

the models is modest (4.2%), and we leave exploration of larger models (>70B) to future work due to compute constraints.

### 4.6 Fine-tuned models have a difficult time verifying False claims

To study cases where fine-tuned models struggle, we analyze Qwen-CLIPPER outputs. Among 1,000 book-level claim pairs in CLIPPER-test, the model fails to verify 37 true claims and 97 false claims, aligning with NoCha findings (Karpinska et al., 2024) that models struggle more with false claims. We investigate perturbations that make false claims appear true and present a taxonomy with examples in Table 4, with further details in §D.2.

## 5 Related work

**Long-context language modeling:** The context size of LLMs has expanded significantly (OpenAI et al., 2024; Dubey et al., 2024; Team et al., 2024; Yang et al., 2025), thanks to position inter- and extrapolation techniques (Press et al., 2022; Su et al., 2023; Peng et al., 2023) and efficient attention implementation (Dao et al., 2022; Dao, 2023; Liu & Abbeel, 2023). Longer data has been used during continual pretraining (Lieber et al., 2024; Xiong et al., 2023) or alignment stage (Bai et al., 2024; Xiong et al., 2024; An et al., 2024). CLIPPER augments existing long-form book texts with synthetic but challenging claims, which serves as the foundation of a fine-tuning pipeline to improve LLMs' understanding and reasoning over long-context data.

**Instruction-tuning data generation:** Short-form data generation methods either induce instruction data from texts (Honovich et al., 2022; Zhou et al., 2023; Li et al., 2024) or generate instruction-output pairs simultaneously (Wang et al., 2023b). Long-context data is synthesized through induction from long-form documents (Pham et al., 2024; Köksal et al., 2023), random document segments (Xiong et al., 2023), or bootstrapping short documents (An et al., 2024; Xu et al., 2024a; Wu et al., 2024a; Wang et al., 2024). CLIPPER uses instruction

induction from compressed document representations to create instruction-tuning data for long-context LLMs.

**Reasoning alignment:** Previous work includes inference-time scaling (OpenAI, 2024; DeepSeek-AI et al., 2025; Muennighoff et al., 2025), prompting (Wei et al., 2023; Kojima et al., 2023; Yao et al., 2023; Wang et al., 2023a), and fine-tuning LLMs on reasoning data (Chung et al., 2022; Huang et al., 2023; Puerto et al., 2024; Yeo et al., 2025). These reasoning data are either human-written rationale (AlKhamissi et al., 2023) or chain of thoughts distilled from larger models (Hsieh et al., 2023; Li et al., 2023; Ho et al., 2023; Zelikman et al., 2022). We find that fine-tuning models on CoTs improves the generated explanations for the validity of book claims.

## 6 Conclusion

We introduce CLIPPER, a compression-based pipeline for generating synthetic narrative claims. We create 19K true/false claims at both book and chapter levels. Our fine-tuned models set a new state-of-the-art among <10B models on claim verification and achieve improvement on narrative understanding tasks. Future work could examine the effect of book-level claims on larger models and explore methods for generating harder claims to better match human-written benchmarks like NoCha.

## Limitations

We only perform hyperparameter tuning on ProLong-Base due to the high cost of the training process. To put things into perspective, training a model on our full test set requires approximately 50 hours on 8 A100 GPUs, each costing $2 per hour to rent. Even training on our tuning subset takes 6 hours. Therefore, extending training further is prohibitively expensive.

Similarly, we do not hire human annotators to write claims for our dataset due to the prohibitive cost and the need for numerous annotators who have thoroughly read the books (Table 8). While this decision may result in less complex claims, our approach offers greater adaptability to new books while significantly reducing costs.

The compression stage can be challenging to fine-tune, subject to model biases, and prone to potential hallucinations. Due to the large volume of data, verifying its accuracy is also difficult, whether via prompting or human annotation. However, we note that prior research has demonstrated that LLMs are capable of producing high-quality summaries of long documents Chang et al. (2024); Kim et al. (2024). In addition, these compressed representations could still provide a strong foundation for claim generations, as most of CLIPPER's claims are grounded in the original book (subsection 2.3).

## Acknowledgment

We extend special gratitude to Marzena Karpinska for benchmarking our models on NoCha and giving us helpful insights into the models' behaviors. This project was partially supported by awards IIS-2046248, IIS-2312949, and IIS-2202506 from the National Science Foundation (NSF).

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

## A    Data Collection

### A.1    Books used in manual analysis

Table 5 lists six books used in our manual analysis. These books are chosen due to the annotator's familiarity with the content, which eases the manual verification process.

| Title | Author | Publication Year | Number of Tokens | Number of Chapters |
|---|---|---|---|---|
| Anne of the Island | L. M. Montgomery | 1915 | 111,337 | 41 |
| Alice in Wonderland | Lewis Carroll | 1865 | 36,691 | 12 |
| The Murder of Roger Ackroyd | Agatha Christie | 1926 | 98,602 | 27 |
| The Picture of Dorian Gray | Oscar Wilde | 1890 | 105368 | 20 |
| Frankenstein | Mary Shelley | 1818 | 97,574 | 24 |
| The Adventures of Tom Sawyer | Mark Twain | 1876 | 97,968 | 35 |

Table 5: Six books used in our manual analysis. Books are chosen due to familiarity with the content.

### A.2    Does memorization have an effect on claim verification performance?

We measure the performance of the models used for fine-tuning on our test set, with and without book text. We provide the book title and author name where the book text is not provided. Our hypothesis is that if the model does better than the random chance baseline (25% accuracy) without the book text, then the claims are either too easy or can be verified without even reasoning over the texts.

| Models | No Text | With Text |
|---|---|---|
| ProLong-Instruct | 0.0% | 35.6% |
| Llama-Instruct | 0.0% | 32.8% |
| Qwen-Instruct | 0.0% | 51.4% |

Table 6: Accuracy on CLIPPER's test set (with and without book texts).

Table 6 shows that all baseline models perform below random chance, significantly trailing behind the performance achieved when the book text is included in the claim verification prompt. These results indicate that even if a model has memorized the book texts or generated claims, such memorization does not affect its performance on the task itself.

### A.3    Are the True/False claims distinguishable without the book texts?

We ask the question of whether distinguishing between True and False claims is inherently too easy. If so, then the high performance of the fine-tuned models may be attributed merely to their ability to detect formatting cues rather than actually reasoning. To investigate this, we prompt both baseline and fine-tuned models to verify claims without providing any book texts or metadata. Our hypothesis is that if a model performs better than random chance under these conditions, then the claims are likely too easily distinguishable based on their formatting alone.

| Models | Before SFT | After SFT |
|---|---|---|
| ProLong-Instruct | 0.0% | 25.2% |
| Llama-Instruct | 20.2% | 13.8% |
| Qwen-Instruct | 21.7% | 22.9% |

Table 7: Accuracy on CLIPPER's test set (no book text or metadata provided).

As shown in Table 7, even after fine-tuning, the models perform only marginally above random guessing. We conclude that, without the contextual information from the book text, True/False claims are not easily distinguishable.

### A.4 Cost Analysis

Table 8 shows the cost incurred by running each stage of our data synthesis pipeline. With the exception of deduplication, which is done by GPT-4o, each stage of the pipeline is performed by Claude. Table 9 shows the estimated per claim cost for the naïve versus main approach based on estimated cost for 6 books. For human annotation, NoCha (Karpinska et al., 2024) reports that their total cost of annotating 1,001 claim pairs is $3,327 USD, so each claim costs around $1.7.

| Stage | Cost |
|---|---|
| Book summary generation | $0.0021 |
| Chapter outline generation | $0.0107 |
| Book-level claim synthesis | $0.0129 |
| Chapter-level claim synthesis | $0.0172 |
| Deduplication | $0.0021 |
| Verification | $0.0064 |
| Total | $0.0514 |

Table 8: Cost to run pipeline per claim (in US dollars, rounded to four decimal places).

| | NAÏVE | CLIPPER |
|---|---|---|
| Cost per claim (book-level) | $0.09 | $0.07 |
| Cost per claim (chap-level) | $0.04 | $0.02 |

Table 9: Estimated cost for our NAÏVE vs CLIPPER approach (rounded to two decimal places)

### A.5 Prompts

Table 10 shows stages to construct CLIPPER, mapped to their corrresponding prompts.

| Prompt | Figure |
|---|---|
| Chapter outline generation | 3 |
| Book summary generation | 4 |
| Chapter-level claim extraction | 5 |
| Book-level claim extraction | 6 |
| Claim deduplication | 7 |
| Claim verification | 8, 9, 10, 11, 12 |
| Chapter-level claim extraction (NAÏVE) | 15 |
| Book-level claim extraction (NAÏVE) | 14 |

Table 10: Figure references for each prompt.

### A.6 Using DeepSeek-Distill to measure CoT groundedness

We evaluate the model on 66 annotated claims from §2.3 and measure its agreement with human annotations (Table 11). Among the models tested, DeepSeek-Distill aligns most closely with human judgments, with only one instance of disagreement, outperforming other models like GPT-4o (10 disagreements) and LLaMA-3.1-70B-Instruct (3 disagreements). Although Llama-70B performs comparably, it fails to provide clear explanations for its decisions and instead generating generic reasoning messages that lack specificity to samples. Therefore, we use DeepSeek-Distill to measure CoT groundedness in our dataset.

---

**Prompt for Chapter Outline Generation**

```
Your task is to create a detailed and objective outline for Chapter {order} of a
 book. You will be provided with the text of Chapter {order}. Ensure that your
 outline faithfully represents the content of the text.

First, carefully read the text of Chapter {order}:
<current_chapter>
{curr}
<current_chapter>

Finally, create an outline for Chapter {order} using the following format:

<synopsis>A one-sentence summary of the current chapter.</synopsis>
<events>A chronological list of at most 7 major events in the chapter. The list
 should formatted as a numbered list. Each event should be one sentence long,
 describing specific details on what happens, where it happens, and which
 characters are involved. DO NOT include subjective interpretation of the events
 .</events>
<characters>A numbered list of characters in the chapter. Include only those that
  are mentioned in the major events. Each character should have the format: [
 character full name]: [character role/relationship with the main characters], [
 physical appearance (if mentioned)], [personality (if mentioned)], first seen at
  [the first setting the character is in], last seen at [last setting the
 character is in].</characters>

Now, create an objective outline for Chapter {order} based on the provided text.
 Ensure that your outline is concise, coherent, and accurately represents the
 content of the chapter.

<synopsis>
[One-sentence summary of the current chapter.]
</synopsis>
<events>
1. [Event 1: Specific details on specific details on what happens, where it
 happens, and which characters are involved.]
2. [Event 2: Specific details on what happens, where it happens, and which
 characters are involved.]
3. [Event 3: Specific details on what happens, where it happens, and which
 characters are involved.]
4. [Event 4: Specific details on what happens, where it happens, and which
 characters are involved.]
5. [Event 5: Specific details on what happens, where it happens, and which
 characters are involved.]
6. [Event 6: Specific details on what happens, where it happens, and which
 characters are involved.]
7. [Event 7: Specific details on what happens, where it happens, and which
 characters are involved.]
</events>
<characters>
1. [Character 1: Character role/relationship with the main characters, physical
 appearance (if mentioned), personality (if mentioned), first seen at the first
 setting the character is in, last seen at the last setting the character is in.]
(Repeat the format for each character mentioned in the major events.)
</characters>

Remember to focus on the objective representation of the chapter content and
 avoid adding personal opinions or interpretations. Good luck!
```

Figure 3: Prompt for generating chapter outlines in our dataset.

---
**Prompt for Generating Book Summary**

```
Your task is to write a summary for the book below, make sure to include vital
 information related to key events, backgrounds, settings, characters, their
 objectives, and motivations. You must briefly introduce characters (with their
 full name), places, and other major elements if they are being mentioned for the
 first time in the summary. The book may feature non-linear narratives,
 flashbacks, switches between alternate worlds or viewpoints, etc. Therefore, you
 should organize the summary so it presents a consistent and chronological
 narrative. The summary must be within 1000 words. The summary should span
 multiple paragraphs and should be written as a single continuous narrative, not
 as a list of bullet points or an outline. DO NOT include the book name in the
 summary.

#Book
{book}

#Summary
```
---

Figure 4: Prompt for generating chapter outlines in our dataset.

| Judge Models | % Agreement |
|---|---|
| GPT-4o | 84.8% |
| Llama-3.1-70b-Instruct | 95.5% |
| DeepSeek-Distill-Llama-70B | 98.5% |

Table 11: Percentage of times LLM judges for chain of thought groundedness agree with our manual annotation over 66 samples in Section 2.3.

## Prompt for Extracting Chapter-level Claims

```
Your task is to create factual statements that incorporate multiple events from
 Chapter {chapter} of a book. These factual statements must be objective and
 specific, grounded in the specific chapter, and consistent with the entire book.
  The included events must be specific. The facts cannot contain any
 interpretations, speculations, or subjective statements. In addition to each
 fact, provide a minimally corrupted version of the fact, which sounds plausible
 but is wrong based on the entire book.
To create valid facts, follow these guidelines step-by-step:
1. Carefully read through the book.
2. In a <brainstorm> section:
   - Identify different events that are strongly related to one another in a
 single chapter.
   - Consider the relationship between these events, but DO NOT include this
 brainstorming in the resulting fact. If a meaningful relationship is found, move
  on to the next step; otherwise, proceed to the next fact.
3. Formulate your facts based on this analysis. Ensure that your facts:
   - Contain a single sentence
   - Are coherent with the entire book
   - Include multiple detailed and strongly related events from a single chapter
  in the book
   - Are self-contained and independent of other facts
   - No part of the fact should be subjective interpretations or speculations
   - Do not contradict or duplicate existing facts
   - Do not contain chapter information (e.g. "In Chapter x, ...")
   - Do not contain quotes from the book.
4. Formulate a minimally corrupted version of each fact.
   - Only one aspect of the fact, such as an atomic detail or the relationship
 between details, should be altered.
   - The corrupted fact should sound plausible but be clearly wrong based on the
  entire book.
   - The corrupted fact should be similar with the original fact in terms of
 length and sentence structure to make it harder to verify solely based on
 surface-level features.
First, read the book:
<book>{book}</book>
Finally, review the existing facts:
<existing_facts>{existing_facts}</existing_facts>
Now, generate as many valid facts as possible based on the provided chapter.
 Return "No meaningful fact" if there is no valid fact. Present your facts in the
  following format:
<facts>
<fact_1>
<brainstorm>[Your brainstorm notes here]</brainstorm>
Fact: [Your fact here]
Fact Reasoning: [Your explanation here (including the chapter associated with
 each event)]
Source: [Chapter involved]
Corrupted Fact: [Your corrupted fact here]
Corrupted Fact Reasoning: [Your explanation here]
</fact_1>
[Continue with additional facts...]
</facts>
Remember to create facts that are objectively valid, coherent with the entire
 book, and demonstrate strong, meaningful relationships between the events from
 Chapter {chapter}. No part of the fact can include subjective interpretations or
  generalizations. The relationship must be meaningful. The included events must
 be SPECIFIC and DETAILED!!!!!
```

Figure 5: Prompt for extracting chapter-level claims

## Prompt for Extracting Book-level Claims

```
Your task is to create factual statements that incorporate multiple events from
 multiple chapters of a book. These factual statements must be objective and
 specific, grounded in the involved chapters, and consistent with the entire book
 . The included events must be specific. The facts cannot contain any
 interpretations, speculations, or subjective statements. In addition to each
 fact, provide a minimally corrupted version of the fact, which sounds plausible
 but is wrong based on the entire book.
To create valid facts, follow these guidelines step-by-step:
1. Carefully read through the book.
2. In a <brainstorm> section:
   - Identify different events that are strongly related to one another in
 multiple chapters.
   - Consider the relationship between these events, but DO NOT include this
 brainstorming in the resulting fact. If a meaningful relationship is found, move
  on to the next step; otherwise, proceed to the next fact.
3. Formulate your facts based on this analysis. Ensure that your facts:
   - Contain a single sentence
   - Are coherent with the entire book
   - Include multiple detailed and strongly related events from multiple
 chapters in the book
   - Are self-contained and independent of other facts
   - No part of the fact should be subjective interpretations or speculations
   - Do not contradict or duplicate existing facts
   - Do not contain chapter information (e.g. "In Chapter x, ...")
   - Do not contain quotes from the book.
4. Formulate a minimally corrupted version of each fact.
   - Only one aspect of the fact, such as an atomic detail or the relationship
 between details, should be altered.
   - The corrupted fact should sound plausible but be clearly wrong based on the
  entire book.
   - The corrupted fact should be similar with the original fact in terms of
 length and sentence structure to make it harder to verify solely based on
 surface-level features.
First, read the book:
<book>{book}</book>
Finally, review the existing facts:
<existing_facts>{existing_facts}</existing_facts>
Now, generate as many valid facts as possible based on the provided book. Return
 "No meaningful fact" if there is no valid fact. Present your facts in the
 following format:
<facts>
<fact_1>
<brainstorm>[Your brainstorm notes here]</brainstorm>
Fact: [Your fact here]
Fact Reasoning: [Your explanation here (including the chapter associated with
 each event)]
Source: [Chapters involved]
Corrupted Fact: [Your corrupted fact here]
Corrupted Fact Reasoning: [Your explanation here]
</fact_1>
[Continue with additional facts...]
</facts>
Remember to create facts that are objectively valid, coherent with the book, and
 demonstrate strong, meaningful relationships between the events from multiple
 chapters. No part of the fact can include subjective interpretations or
 generalizations. The relationship must be meaningful. The included events must
 be SPECIFIC and DETAILED!!!!!
```

Figure 6: Prompt for extracting book-level claims

## Prompt for Deduplicating Claims

```
You will be given a list of facts. Your task is to identify all duplicate facts
 within this list. A fact is considered a duplicate if it is exactly the same as
 another fact, or if it contains the same information as another fact, even if
 worded differently.

Here is the list of facts:

<fact_list>
{fact_list}
</fact_list>

To identify the duplicate facts, follow these guidelines step by step:
1. Read through the list of facts carefully.
2. Identify any facts that are exact duplicates of each other, or that convey the
   same atomic information using different wording.
3. Return a list of facts that are duplicates of one other, along with an
 explanation of why they are duplicates.
4. DO NOT return facts that are not duplicates of any other fact in the list.

Here's an example of how your output should look:

<example>
<example_fact_list>
1. Jim worked hard, so he got a promotion.
2. Jim's hard work paid off, and he was promoted.
3. Jim and Sarah worked together on the project.
4. Jim worked hard, so he received praise from his boss, who promoted him.
</example_fact_list>

<example_answer>
- 1, 2: These two facts convey the same atomic information but are worded
 differently.
</example_answer>
</example>

Remember, your goal is to identify all duplicate facts, whether they are exact
 matches or convey the same information in different words. Be thorough in your
 analysis and clear in your explanations. If there is no duplication in the list,
  output "No duplicates found."

<answer>
- [Index of duplicate facts, separated by commas]: [Explanation of why they are
 duplicates]
- [Index of duplicate facts, separated by commas]: [Explanation of why they are
 duplicates]
(... and so on)
</answer>
```

Figure 7: Prompt for de-duplicating claims

---

**Prompt for Verifying Claims with GPT-4o (Part 1)**

```
You will receive a book summary, a chapter outline, and a claim extracted from
 that outline. Your task is to verify whether the claim contains detailed
 information, presents a meaningful relationship, and shows consistency with both
  the book summary and chapter outline.

To verify the claim, follow these steps:
1. Read the summary, outline, and claim carefully to understand the context.
2. Decompose the claim into atomic parts.
3. Analyze each atomic part:
    a. Is the part grounded in the events?
    b. Does this part contradict any information in the summary or outline? Keep
 in mind that some books may have discontinuous plots or events, so just because
 a detail is mentioned before or after another in the summary does not mean they
 are temporally related.
4. Evaluate the relationship between the atomic parts:
    a. Is the relationship objectively valid and meaningful?
    b. Is the relationship a subjective interpretation or assumption not
 explicitly stated in the summary or outline?
    c. Does the relationship make sense based on the book summary and chapter
 outline?
5. Based on your analysis, provide your reasoning and verification result. Your
 reasoning should explain why you believe the claim is or is not valid based on
 the information provided in the book summary and chapter outline.

First, read the book summary:
<book_summary>
{book_summary}
</book_summary>

Next, review the chapter outline:
<chapter_outline>
{chapter_outline}
</chapter_outline>

Finally, consider the following claim:
<claim>
{claim}
</claim>

Here are two examples of valid and invalid claims:
```

Figure 8: Prompt for verifying claims with GPT-4o (Part 1)

```
Prompt for Verifying Claims with GPT-4o (Part 2)

<example_1>
<example_summary>
Laura Hand, Daniel Knowe, and Mo Gorch mysteriously return from a realm of death
 with altered memories, orchestrated by their enigmatic music teacher, Mr. Anabin
 , and the sinister Bogomil. As they grapple with their new realities, including
 Mo's discovery of his grandmother's death and Daniel's complex feelings for
 Laura's sister, Susannah, they face trials set by Anabin and Bogomil to remain
 in the living world. Alongside Bowie, another returned soul, they uncover the
 truth about their deaths while dealing with eerie supernatural encounters. Laura
 's newfound magical abilities strain her relationship with Susannah, leading to
 increasing tensions as Susannah begins to remember the truth.
As the trio navigates their altered lives, they become entangled in a larger,
 dangerous game orchestrated by Malo Mogge, Anabin, and Bogomil, who guard the
 door between life and death. Mo and Susannah discover that a Harmony guitar,
 hidden by Susannah, is the key sought by Malo Mogge, a powerful entity seeking
 immense power. The story culminates in a chaotic battle in Lovesend, where Laura
 , consumed by grief, vows to kill Malo Mogge. After absorbing Mogge's magic and
 becoming a powerful goddess, Laura faces the challenge of balancing her divine
 powers with her passion for music, while the other characters embrace their new
 roles.
</example_summary>

<example_chapter_outline>
## Chapter 15 outline
1. Susannah gets frustrated about Laura and Daniel being close to each other and
 smashes Laura's old guitar in a fit of anger and frustration.
2. Mo eats a breakfast casserole made by Jenny and buys doughnuts and bagels on
 his way to Laura's house, feeling a mix of hunger and sadness.
3. Mo arrives at the Hands' house, where Laura and Daniel are eating ramen to
 satisfy their unusual hunger.
4. Daniel reveals that he and Laura have swapped ears due to Mr. Anabin's magic,
 and they discuss the implications of this mistake.
5. Mo shares his encounter with a mysterious figure outside his house, leading to
  a heated discussion about Bogomil and Mr. Anabin.
6. The trio creates a list of goals to navigate their situation, including
 staying alive, figuring out how they died, and learning to do magic.
7. They attempt to perform magic by trying to transform a saltshaker into a
 hairless cat but fail, leading to further frustration.
8. Mo leaves the Hands' house, and they discover that the entire yard and house
 are covered in thousands of moths, adding to the surreal nature of their
 situation.
9. Laura finds the broken guitar pieces in her room, causing confusion and
 suspicion among the trio.
</example_chapter_outline>

<example_claim_1>
Laura finds the broken guitar pieces in her room, which Susannah smashed in a fit
  of anger and frustration.
</example_claim_1>

<example_verification_1>
<reasoning>
1. Analysis of each atomic part:
    a. Laura finds the broken guitar pieces in her room: This part is grounded in
  event 9 of Chapter 15.
    b. Susannah smashed the guitar in a fit of anger and frustration: This part
 is grounded in event 1 of Chapter 15.
2. Analysis of the relationship between atomic parts:
    a. The relationship is temporal, and thus valid.
```

Figure 9: Prompt for verifying claims with GPT-4o (Part 2)

---

**Prompt for Verifying Claims with GPT-4o (Part 3)**

```
    b. The relationship is explicit and not a subjective interpretation.
    c. The relationship is grounded in the chapter outline. There is no
 contradicting information in the summary.
Since all considerations are satisfied, the claim is VALID.
</reasoning>
<result>
VALID
</result>
</example_verification_1>

<example_claim_2>
Mo shares his encounter with a mysterious figure because Laura and Daniel discuss
  the implications of their swapped hands due to Mr. Anabin's magic.
</example_claim_2>

<example_verification_2>
<reasoning>
1. Analysis of atomic parts:
    a. Mo shares his encounter with a mysterious figure: This part is grounded in
  event 5 of Chapter 15.
    b. Laura and Daniel discuss the implications of their swapped hands due to Mr.
 Anabin's magic: There is no mention of hands being swapped. Even though event 4
  discusses the swapping of ears, it does not relate to hands.
2. Analysis of the relationship between atomic parts:
    a. The relationship is NOT VALID because there is no direct link between Mo
 sharing his encounter and Laura and Daniel discussing the implications of their
 swapped hands.
    b. The relationship is a subjective interpretation and not explicitly
 grounded in the summary or outline.
    c. The relationship is grounded in the chapter outline. There is no
 contradicting information in the summary.
Since 1a., 2a., and 2b. are not satisfied, the claim is INVALID.
</reasoning>
<result>
INVALID
</result>
</example_verification_2>
```

Figure 10: Prompt for verifying claims with GPT-4o (Part 3)

---

**Prompt for Verifying Claims with GPT-4o (Part 4)**

```
<example_claim_3>
Jenny cooks breakfast for Mo because he feels a mix of hunger and sadness.
</example_claim_3>

<example_verification_3>
<reasoning>
1. Analysis of atomic parts:
   a. Jenny cooks breakfast for Mo: This part is grounded in event 2 of Chapter
 15.
   b. Mo feels a mix of hunger and sadness: This part is grounded in event 2 of
 Chapter 15.
2. Analysis of the relationship between atomic parts:
   a. The relationship is INVALID. There is no indication that Jenny cooked
 breakfast for Mo because he felt a mix of hunger and sadness. The events are
 happening simultaneously but are not causally connected.
   b. The relationship is a subjective interpretation and not explicitly
 grounded in the summary or outline.
   c. The relationship is grounded in the chapter outline. There is no
 contradicting information in the summary.
Since 2a. and 2b. are not satisfied, the claim is INVALID.
</reasoning>
<result>
INVALID
</result>
</example_verification_3>
</example_1>

<example_2>
<example_summary>
Sarah Lee discovers an ancient wooden box with strange symbols in her attic,
 which contains a journal revealing the history of a secret society and a
 prophecy about the return of a powerful being known as "The Shadow." As unusual
 events begin to plague her town, Sarah, along with her friend Mark, uncovers
 clues that connect these occurrences to the prophecy. They find a hidden chamber
  beneath the town containing ancient texts and artifacts, including a weapon
 capable of banishing "The Shadow." With this weapon, they confront a member of
 the secret society who attempts to summon "The Shadow," and after a tense battle
 , Sarah successfully uses the weapon to banish the being, restoring peace to the
  town.

Throughout the story, the connection between the ancient artifact, the journal,
 and the unfolding events reveals the central role of the wooden box and the
 secret society in the impending danger. Sarah and Mark's journey highlights
 their struggle to protect their town from supernatural forces while deciphering
 the mysterious symbols and prophecies tied to the powerful entity, "The Shadow."
</example_summary>

<example_chapter_outline>
## Chapter 3 outline
```

Figure 11: Prompt for verifying claims with GPT-4o (Part 4)

---

**Prompt for Verifying Claims with GPT-4o (Part 5)**

```
1. Sarah discovers an ancient artifact in her attic, an intricately carved wooden
   box with strange symbols.
2. She finds an old journal in the box, detailing the history of a secret society
   that once protected the town.
3. The journal reveals a prophecy about the return of a powerful being known as "
   The Shadow."
4. Sarah decides to keep the discovery to herself, fearing that revealing it
   would cause panic.

## Chapter 8 outline
1. Sarah begins to notice strange occurrences around town, like unusual weather
   patterns and eerie shadows.
2. She consults the journal again and discovers a passage that seems to describe
   unusual event patterns as signs of "The Shadow's" return.
3. Sarah's friend Mark, who is a local historian, suggests that they investigate
   further by visiting the town's library.
4. At the library, they find more texts related to the secret society and "The
   Shadow."
5. Mark went home to rest after a long day, where he met his mother.
<example_chapter_outline>

<example_claim>
Sarah decides to keep the discovery to herself, which reveals the challenges in
 Mark and Sarah's friendship.
</example_claim>

<example_verification>
<reasoning>
1. Analysis of atomic parts:
   a. Sarah decides to keep the discovery to herself: This part is grounded in
 event 4 of Chapter 3.
   b. "reveals the challenges in Mark and Sarah's friendship": This part is a
 subjective interpretation and not explicitly grounded in the summary or outline.
2. Analysis of the relationship between atomic parts:
   a. The relationship is not meaningful, as there is no direct connection
 between Sarah keeping the discovery to herself and revealing challenges in Mark
 and Sarah's friendship.
   b. The relationship is a subjective interpretation and not explicitly
 grounded in the summary or outline.
   c. The relationship does not contradict any information in the summary or
 outline.
Since 1b., 2a., and 2b. are not satisfied, the claim is INVALID.
</reasoning>
<result>
INVALID
</result>

<example_claim>
Sarah finds an old journal in the room, which prompted Mark to go home and meet
 his mother.
</example_claim>
<example_verification>
<reasoning>
1. Analysis of atomic parts:
   a. Sarah finds an old journal in the room: This part is grounded in event 2
 of Chapter 3.
   b. Mark went home to rest after a long day, where he met his mother: This
 part is grounded in event 5 of Chapter 8.
```

Figure 12: Prompt for verifying claims with GPT-4o (Part 5)

---

**Prompt for Verifying Claims with GPT-4o (Part 6)**

```
2. Analysis of the relationship between atomic parts:
    a. The relationship is not meaningful, as there is no direct connection
 between Sarah finding the journal and Mark going home to meet his mother.
    b. The relationship is a subjective interpretation and not explicitly
 grounded in the summary or outline.
    c. The relationship does not contradict any information in the summary or
 outline.
Since 2a. and 2b. are not satisfied, the claim is INVALID.
</reasoning>
<result>
INVALID
</result>
</example_verification>
</example_2>

Now, it's your turn to verify the claim based on the provided book summary,
 chapter outline, and claim. Present your response in the following format:
<verification>
<reasoning>
[Provide your detailed reasoning here, explaining why the claim is or is not
 meaningful and coherent with the book summary and chapter outline.]
</reasoning>
<result>
[State whether the claim is VALID or INVALID. Use VALID if the claim portrays a
 meaningful relationship and is coherent with the book summary and chapter
 outline. Use INVALID if it does not.]
</result>
</verification>

Remember to base your verification solely on the information provided in the book
  summary, chapter outline, and the claim itself. Verify that the relationship
 makes sense and is objectively valid. Do not introduce external information or
 make assumptions beyond what is given.
```

---

Figure 13: Prompt for verifying claims with GPT-4o (Part 6)

---

**Prompt for Generating Book-level Claims in NAÏVE**

```
Your task is to create factual statements that incorporate multiple events from
 multiple chapters of a book. These factual statements must be objective and
 specific, grounded in the involved chapters, and consistent with the entire book
 . The included events must be specific. The facts cannot contain any
 interpretations, speculations, or subjective statements. In addition to each
 fact, provide a minimally corrupted version of the fact, which sounds plausible
 but is wrong based on the entire book.
To create valid facts, follow these guidelines step-by-step:
1. Carefully read through the book.
2. In a <brainstorm> section:
   - Identify different events that are strongly related to one another in
 multiple chapters.
   - Consider the relationship between these events, but DO NOT include this
 brainstorming in the resulting fact. If a meaningful relationship is found, move
  on to the next step; otherwise, proceed to the next fact.
3. Formulate your facts based on this analysis. Ensure that your facts:
   - Contain a single sentence
   - Are coherent with the entire book
   - Include multiple detailed and strongly related events from multiple
 chapters in the book
   - Are self-contained and independent of other facts
   - No part of the fact should be subjective interpretations or speculations
   - Do not contradict or duplicate existing facts
   - Do not contain chapter information (e.g. "In Chapter x, ...")
   - Do not contain quotes from the book.
4. Formulate a minimally corrupted version of each fact.
   - Only one aspect of the fact, such as an atomic detail or the relationship
 between details, should be altered.
   - The corrupted fact should sound plausible but be clearly wrong based on the
  entire book.
   - The corrupted fact should be similar with the original fact in terms of
 length and sentence structure to make it harder to verify solely based on
 surface-level features.
First, read the book:
<book>{book}</book>
Finally, review the existing facts:
<existing_facts>{existing_facts}</existing_facts>

Now, generate as many valid facts as possible based on the provided book. Return
 "No meaningful fact" if there is no valid fact. Present your facts in the
 following format:

<facts>
<fact_1>
<brainstorm>[Your brainstorm notes here]</brainstorm>
Fact: [Your fact here]
Fact Reasoning: [Your explanation here (including the chapter associated with
 each event)]
Source: [Chapters involved]
Corrupted Fact: [Your corrupted fact here]
Corrupted Fact Reasoning: [Your explanation here]
</fact_1>
[Continue with additional facts...]
</facts>
Remember to create facts that are objectively valid, coherent with the book, and
 demonstrate strong, meaningful relationships between the events from multiple
 chapters. No part of the fact can include subjective interpretations or
 generalizations. The relationship must be meaningful. The included events must
 be SPECIFIC and DETAILED!!!!!
```

Figure 14: Prompt for generating book-level claims in NAÏVE.

---

**Prompt for Generating Chapter-level Claims in NAÏVE**

```
Your task is to create factual statements that incorporate multiple events from
 Chapter {chapter} of a book. These factual statements must be objective and
 specific, grounded in the specific chapter, and consistent with the entire book.
  The included events must be specific. The facts cannot contain any
 interpretations, speculations, or subjective statements. In addition to each
 fact, provide a minimally corrupted version of the fact, which sounds plausible
 but is wrong based on the entire book.
To create valid facts, follow these guidelines step-by-step:
1. Carefully read through the book.
2. In a <brainstorm> section:
   - Identify different events that are strongly related to one another in a
 single chapter.
   - Consider the relationship between these events, but DO NOT include this
 brainstorming in the resulting fact. If a meaningful relationship is found, move
  on to the next step; otherwise, proceed to the next fact.
3. Formulate your facts based on this analysis. Ensure that your facts:
   - Contain a single sentence
   - Are coherent with the entire book
   - Include multiple detailed and strongly related events from a single chapter
  in the book
   - Are self-contained and independent of other facts
   - No part of the fact should be subjective interpretations or speculations
   - Do not contradict or duplicate existing facts
   - Do not contain chapter information (e.g. "In Chapter x, ...")
   - Do not contain quotes from the book.
4. Formulate a minimally corrupted version of each fact.
   - Only one aspect of the fact, such as an atomic detail or the relationship
 between details, should be altered.
   - The corrupted fact should sound plausible but be clearly wrong based on the
  entire book.
   - The corrupted fact should be similar with the original fact in terms of
 length and sentence structure to make it harder to verify solely based on
 surface-level features.
First, read the book:
<book>{book}</book>
Finally, review the existing facts:
<existing_facts>{existing_facts}</existing_facts>
Now, generate as many valid facts as possible based on the provided chapter.
 Return "No meaningful fact" if there is no valid fact. Present your facts in the
  following format:
<facts>
<fact_1>
<brainstorm>[Your brainstorm notes here]</brainstorm>
Fact: [Your fact here]
Fact Reasoning: [Your explanation here (including the chapter associated with
 each event)]
Source: [Chapter involved]
Corrupted Fact: [Your corrupted fact here]
Corrupted Fact Reasoning: [Your explanation here]
</fact_1>
[Continue with additional facts...]
</facts>
Remember to create facts that are objectively valid, coherent with the entire
 book, and demonstrate strong, meaningful relationships between the events from
 Chapter {chapter}. No part of the fact can include subjective interpretations or
  generalizations. The relationship must be meaningful. The included events must
 be SPECIFIC and DETAILED!!!!!
```

Figure 15: Prompt for generating chapter-level claims in NAÏVE.

## B Training

### B.1 Codebases

To fine-tune models in the Llama family, we adopt the ProLong codebase,[17] which integrates PyTorch (Paszke et al., 2019) and Hugging Face (Wolf et al., 2020) for model training, FlashAttention-2 (Dao, 2023) for efficient attention computation, and DeepSpeed Ulysses (Jacobs et al., 2023) for sequence parallelism, enabling training across 8 A100 GPUs. For fine-tuning Qwen-Instruct (Qwen-Instruct), we use the 360-LlamaFactory codebase,[18] a modification of Llama-Factory[19] that incorporates sequence parallelism via zigzag ring attention (Liu & Abbeel, 2023; Liu et al., 2023). We choose ProLong-Base (ProLong-base) over ProLong-Instruct (ProLong-instruct) based on a small fine-tuning experiment, where we fine-tune both ProLong-instruct and ProLong-base on 2K training examples. This experiment shows that ProLong-base outperforms ProLong-instruct by 61.6% and 59.7%, respectively.

### B.2 Hyperparameter Tuning

Table 12 summarizes the performance of each configuration from our hyperparameter tuning experiment on 100 samples from CLIPPER's dev set.

| Learning Rate | Batch Size | Dev Set Accuracy |
|:---:|:---:|:---:|
| 1e-5 | 16 | 26% |
| 1e-6 | 16 | **74**% |
| 1e-7 | 16 | 71% |
| 1e-5 | 32 | 34% |
| 1e-6 | 32 | 73% |
| 1e-7 | 32 | 69% |

Table 12: Hyperparameter tuning results. Each model is fine-tuned for 1 epoch and tested on a subset of 100 samples from our dev set.

## C Evaluation

### C.1 Configuration for ∞Bench QA Evaluation

In HELMET (Yen et al., 2024), for the ∞ench QA task, the default configuration sets the output maximum length to 10 tokens and uses ROUGE F1 (Lin, 2004) as the evaluation metric. Upon closer examination of the outputs from both models, we identify critical flaws in the default setup. These findings eventually motivate us to remove the maximum length restriction and adopt the LLM-as-a-judge evaluation approach using GPT-4o as the judge. Below, we provide more details on our analysis.

**Setting max output length to 10 tokens frequently cause truncations:** In Table 13, we show the taxonomy we derive from our analysis. Here, we define a truncation to be when a model's response is heavily cut off, *making it impossible to determine the correctness of the output*. Out of all 100 evaluated examples, Qwen2.5-7B-Instruct's outputs get truncated 24 times, while Qwen-CLIPPER's outputs get truncated 45 times. After removing the 10-token maximum length restriction[20], we observe that 9 of the 24 previously truncated outputs from Qwen-Instructshould be counted as correct. For Qwen-CLIPPER, this correction is even

---

[17]https://github.com/princeton-nlp/ProLong
[18]https://github.com/Qihoo360/360-LLaMA-Factory
[19]https://github.com/hiyouga/LLaMA-Factory
[20]Without the maximum length limit, Qwen2.5-7B-Instruct's outputs are on average 20.9 tokens long, and Qwen-CLIPPER's outputs have 25.8 tokens on average.

## Finetuning Prompt

*System Message*

You are an expert at verifying claims from fictional narratives.

*User Message*

You are provided with a context and a statement. Your task is to carefully read the context and then determine whether the statement is TRUE or FALSE.

Answer TRUE if the statement is true in its entirety based on the context provided.
Answer FALSE if any part of the statement is false based on the context provided.

<context>**{book_text}**</context>
<statement>**{claim_text}**</statement>

<question>Based on the context provided, is the above statement TRUE or FALSE?
</question>

First provide an explanation of your decision-making process, and then provide your final answer. Use the following format:

<explanation>YOUR EXPLANATION</explanation>
<answer>YOUR ANSWER</answer>

*Assistant Message*

<explanation>Here are the relevant detail from the text:
- *Chapter x: …*
- *Chapter y: …*
These chapters have relationship *z*, which makes the claim TRUE/FALSE.</explanation>

<answer>TRUE/FALSE</answer>

Figure 16: How we structure our fine-tuning prompts, which include system message, user message, and assistant message. Placeholders (colored in **light cayenne**) will be replaced with actual text from the dataset. The contents between <context>, <explanation>, and <answer> tags are generated (Section 2.3).

| CATEGORY | DEFINITION | EXAMPLE QUESTION | EXAMPLE GROUND-TRUTH | EXAMPLE MODEL ANSWER |
|---|---|---|---|---|
| **Cases that should be counted as incorrect** | | | | |
| Wrong | The model's answer is evidently wrong. | What is the home Edna moves into in New Orleans called? | The pigeon house | The text does not provide information about Edna moving |
| **Cases that should be counted as correct** | | | | |
| Full match | The model's answer perfectly matches the ground-truth answer. | Which among Annalisa, Seb, Peyton, and Gannonmarie is not Mrs. Bronwyn's child? | Peyton | Peyton |
| Correct (Phrasing) | The model's answer is correct, but it has a different phrasing than the ground-truth answer. | How old is Felicity at the start of his narration? | Thirty | Felicity is 30 years old at |
| Correct (Long) | The model's answer is correct, but it is longer than the ground-truth. | What kind of pet does Madame Bowen keep? | a cat | Madame Bowen keeps a cat as a pet. |
| Correct (Short) | The model's answer is correct, but it is shorter than the ground-truth. | Why is Tasha Teigan out of jail? | He has been released on parole. | Paroled. |
| **Cases where correctness is ambiguous** | | | | |
| Truncation | The model's answer has been heavily truncated, making it impossible to tell the correctness of the answer. | What is to be built in place of the Lars home on Wickham Place? | Flats | The house on Wickham Place is to be replaced |

Table 13: Taxonomy from our analysis on the ∞Bench QA outputs of Qwen2.5-7B-Instruct and Qwen-CLIPPER. Example model outputs are from Qwen-CLIPPER except the one for "Correct (Short)", which is from Qwen2.5-7B-Instruct (all generated under the default setup where maximum output tokens is set to 10).

| CATEGORY | QWEN-INST | QWEN-BC |
|---|---|---|
| **Cases that should be counted as incorrect** | | |
| Wrong | 41 | 26 |
| **Cases that should be counted as correct** | | |
| Full match | 17 | 4 |
| Correct (Phrasing) | 3 | 7 |
| Correct (Long) | 13 | 18 |
| Correct (Short) | 2 | 0 |
| **Cases where correctness is ambiguous** | | |
| Truncation | 24 | 45 |

Table 14: Raw counts of taxonomy categories for Qwen-Instruct and Qwen-CLIPPER, with outputs generated using the default maximum length of 10 tokens.

more significant, with 25 of the 45 truncated outputs being technically correct. We combine these numbers with numbers from the four rows in Table 14 that indicate correctness, and find that Qwen2.5-7B-Instruct has an overall accuracy of 44%, while Qwen-CLIPPER has 54%.

**ROUGE F1 is not a reliable metric:** If we use ROUGE F1 as the metric, Qwen2.5-7B-Instruct achieves a score of 27.4, while Qwen-CLIPPERachieves a score of 18.0. This result sharply contrasts with the accuracies we obtain in the preceding paragraph, and does not

| QUESTION | GROUND-TRUTH | MODEL ANSWER | ROUGE F1 | EXPLANATION |
|---|---|---|---|---|
| How old is Felicity at the start of his narration? | Thirty | Felicity is 30 years old at | 0 | The model is correct, but it uses the numerical form of the number. |
| What gender does Harris predict Cal will be? | MALE | Harris predicts that Cal will be a boy. | 0 | The model is correct, but it phrases it differently, resulting in no word overlap. |
| When is Jarod's birthday? | NOVEMBER 9 | Jarod's birthday is on November 16 | 0.22 | The model is completely wrong, but it gets the same score as the model answer in the row below, which contains a correct answer. |
| In which state is Gopher Prairie located? | Minnesota | Gopher Prairie is located in Minnesota. This is | 0.22 | The model is correct, but it gets the same score as the wrong model answer above, just because the output is much longer than the ground-truth. |

Table 15: Examples showing that ROUGE-F1 is an unreliable metric.

reflect the actual performance of the models. Lots of prior work have shown that ROUGE correlates poorly with human judgment (Goyal et al., 2023; Chang et al., 2024). Our manual analysis reveals that this metric is overly sensitive to length, and does not capture the correctness of the model outputs. We show several examples in Table 15.

## D   Results

### D.1   Impact of chapter distance and book length on test set performance

Figure 17 shows that test set accuracy peaks when the distance between chapters in a claim is around 40–60K tokens (roughly the midpoint of a book). When that gap shrinks below or stretches beyond 60K tokens, performance dips by about 10%, leaving no definitive pattern beyond this sweet spot.

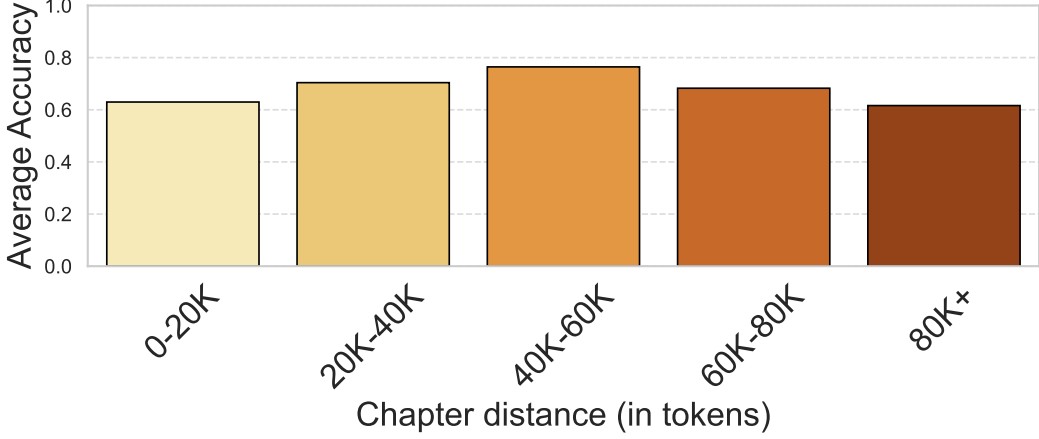

Figure 17: Accuracy of CLIPPER-Prolong-balanced on the test set (book-level claims), grouped by the distance (in tokens) between source events in each claim.

We also find that overall book length does not strongly influence accuracy, except in cases where the text exceeds 110K tokens. In these longer works, accuracy is about 5% higher than in shorter books, as shown in Figure 18. While this slight edge may hint at advantages

in more expansive narratives, the model's broader performance remains steady across most book lengths.

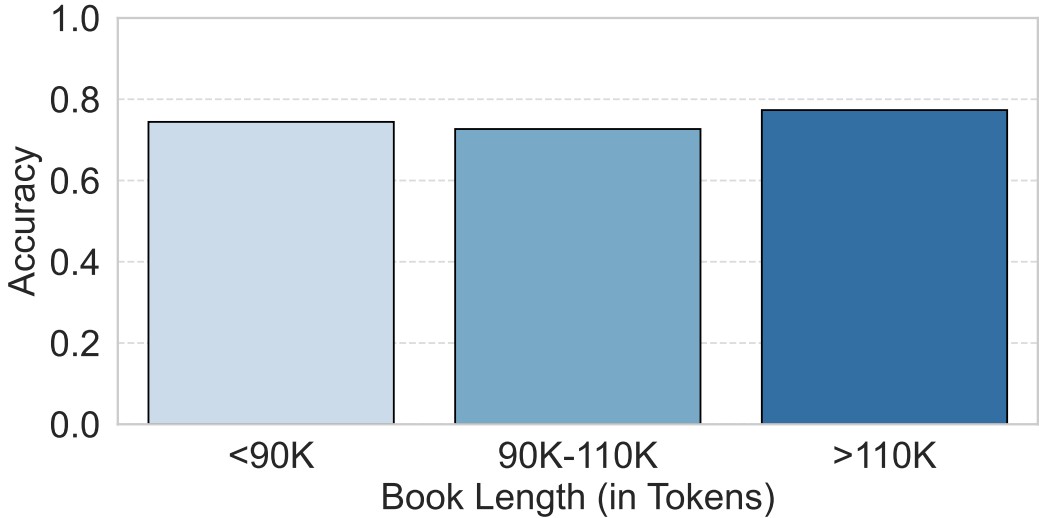

Figure 18: ProLong-CLIPPER's performance on test set, grouped by the number of tokens in each book.

We finally examine the possible effect of event placement on ProLong-CLIPPER's performance on the test set. Interestingly, there is no strong "lost-in-the-middle" effect regarding event placement in the book (Liu et al., 2024). As shown in Figure 19, accuracy is usually the highest when the claim involves events that appear at the beginning (0-0.4, around 82%) rather than at the end of the book (0.8-1, around 78%).

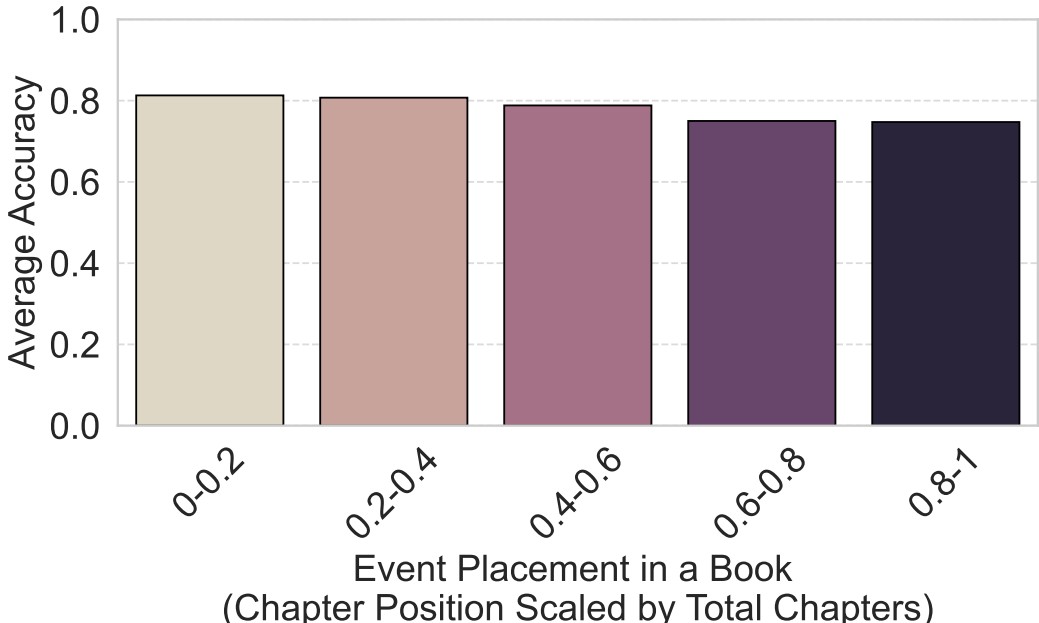

Figure 19: Accuracy of ProLong-CLIPPER on the test set (chapter-level claims), grouped by the event placement in the book (0-0.2 includes events are at the beginning, while 0.8-1 includes events towards the end).

### D.2 False claim error analysis

In Table 16, we provide detailed definitions for each category from the false claim error analysis in §4.6. To explore instances where fine-tuned models still struggle, we conduct an in-depth analysis of Qwen-CLIPPER outputs. Of the 1,000 book-level claims in the test set, the model fails to verify 37 true claims and 97 false claims. This pattern is consistent with findings from NoCha (Karpinska et al., 2024), which highlight that models tend to have greater difficulty verifying false claims. Notably, in 95 cases, the model successfully validates the true claim but fails to validate the corresponding false claim. This raises an important question: *What specific perturbations make a false claim appear true to the model?*

Through careful manual analysis, we derive a taxonomy of such perturbations and present them in Table 4. The most frequent perturbations are changes to events (43.2%) and people (31.6%), such as altering actions or misattributing roles. Less frequent but notable are modifications to objects (15.8%), locations (13.7%), time (6.3%), and affect (4.2%). All these perturbations introduce plausible-sounding variations that the model may struggle to detect without fully understanding the narrative.

| CATEGORY | DEFINITION |
|---|---|
| Event | Refers to the alteration or misrepresentation of the actions, occurrences, or processes described in a claim. |
| Person | Involves substituting or misattributing individuals involved in a claim. |
| Object | Concerns the manipulation or substitution of physical items or artifacts mentioned in a claim. |
| Location | Relates to changing or misrepresenting the places where events occur. |
| Time | Pertains to the sequencing or timing of events being distorted or swapped. |
| Affect | Deals with altering the emotional state, attitude, or disposition described in a claim. |

Table 16: Definitions for each category of perturbations that cause a false claim to be misclassified as true in the error analysis in §4.6.

### D.3 Full results on LM Harness and HELMET

Table 17 shows the results of all models on popular short-form benchmarks. Overall, our fine-tuned models, especially Qwen-CLIPPER, do not degrade that significantly from the baseline models even though it has been fine-tuned on longer data. Table 18 shows the results of HELMET on recall, RAG, passage re-ranking, and retrieval tasks. Overall, fine-tuned models achieve synthetic recall and RAG scores comparable to the baseline models, while generally delivering improved re-ranking and more robust ICL performance.

| Models | IFEval | BBH | Math lvl5 | GPQA | MMLU-Pro | Arc-Challenge | GSM8K | HellaSwag | WinoGrande |
|---|---|---|---|---|---|---|---|---|---|
| Llama-Instruct | **59.35** | 50.93 | 12.81 | 31.96 | 37.77 | 51.54 | 75.06 | 59.05 | 74.19 |
| Qwen-Instruct | 54.00 | 54.60 | **24.80** | 33.40 | 43.80 | 53.20 | 77.70 | 61.80 | 69.20 |
| Prolong-instruct-noft | 58.87 | 49.86 | 5.28 | 29.35 | 32.43 | **58.36** | 68.06 | **80.75** | **74.43** |
| Qwen-CLIPPER | 50.65 | **55.50** | 22.51 | **33.82** | **44.49** | 53.84 | **78.32** | 61.71 | 69.06 |
| ProLong-CLIPPER | 7.91 | 48.03 | 5.42 | 27.88 | 32.35 | 50.68 | 60.80 | 60.44 | 73.24 |
| Llama-CLIPPER | 45.43 | 50.02 | 12.77 | 30.59 | 37.55 | 53.50 | 74.91 | 78.65 | 73.40 |
| ProLong-WritingPrompts | 11.75 | 47.32 | 3.59 | 30.73 | 26.29 | 50.51 | 39.04 | 76.36 | 70.64 |
| ProLong-CLIPPER-book | 6.39 | 49.31 | 5.53 | 29.47 | 32.38 | 54.78 | 62.02 | 79.27 | 72.93 |
| ProLong-CLIPPER-chapter | 4.59 | 49.64 | 5.63 | 29.77 | 32.35 | 54.27 | 61.22 | 79.14 | 74.27 |

Table 17: Performance on popular short-form benchmarks (evaluated using Language Model Evaluation Harness).

### D.4 Performance of ProLong-Base on claim verification and narrative understanding benchmarks

Table 19 shows accuracy of ProLong-Baseon long-context reasoning and narrative understanding benchmarks. Even though ProLong-Base's test set performance is much worse

| Model | Synthetic Recall (Ruler) | | | | RAG | | | Re-ranking | ICL | | | | |
|---|---|---|---|---|---|---|---|---|---|---|---|---|---|
| | niah_mk_2 | recall | niah_mk_3 | recall | niah_mv | recall | json_kv | nqh | triviaqa | hotpotqa | msmarco | trec_coarse | trec_fine |
| Llama-Instruct | 98 | 88 | 78.75 | 96 | 48.17 | 80.67 | 56 | 13.66 | 73 | 72 | 91 | 91 | 88 |
| Qwen-Instruct | **100.0** | 98.0 | **83.3** | 98.8 | 20.3 | 47.3 | 24.0 | 0 | 78.0 | 20.0 | 6.0 | 11.0 | 7.0 |
| ProLong-Instruct | 98.0 | 98.0 | 46.8 | 99.3 | **54.3** | 91.3 | **57.7** | 25.0 | 86.0 | 59.0 | **92.0** | 94.0 | 89.0 |
| ProLong-CLIPPER | 99 | 92 | 27.75 | 99 | 52.5 | **92** | 51.33 | **27.50** | 92 | 72 | 90 | 94 | **90** |
| Qwen-CLIPPER | 81 | 45 | 45 | 45 | 38.16 | 66.5 | 36 | 3.17 | 73 | 52 | 87 | 78 | |
| Llama-CLIPPER | 98 | **99** | 26 | **100** | 48.5 | 85.17 | 55.67 | 22.90 | 87 | **81** | 90 | **95** | 86 |
| ProLong-WritingPrompts | 99 | 87 | 31.75 | **100** | 50.5 | 89.17 | 51.33 | 18.30 | 91 | 65 | 91 | **95** | 88 |

Table 18: Performance on HELMET for recall, RAG, passage re-ranking, and retrieval tasks. Fine-tuned models achieve synthetic recall and RAG scores comparable to the baseline models, while generally delivering improved re-ranking and more robust ICL performance.

than ProLong-Instruct, performance on other narrative understanding tasks is comparable between the two models.

| CLIPPER-Test | NarrativeQA | MuSR | ∞Bench QA |
|---|---|---|---|
| 23.9% | 46.0% | 39.8% | 42.5% |

Table 19: Performance of ProLong-Base on long-context reasoning and narrative understanding benchmarks. Even though ProLong-Base's test set performance is much worse than ProLong-Instruct, performance on other narrative understanding tasks is comparable between the two models.

| | ProLong-CP-book | ProLong-CP-chap |
|---|---|---|
| Test-book | 74.8% | 78.2% |
| Test-chapter | 75.2% | 80.2% |
| Overall | 75.0% | 79.2% |

Table 20: Test set performance of models trained exclusively on either book-level claims or chapter-level claims, with accuracy measured for book-level, chapter-level, and overall claims. CP stands for CLIPPER.

## D.5 Performance of retrieval-augmented baselines on CLIPPER-test

To better isolate the contribution of CLIPPER, we present the performance of a retrieval-augmented approach on CLIPPER-test. Specifically, we report results of an approach where we use BM25 (Robertson et al., 1995) to retrieve the top 50 relevant book passages (each no longer than 256 words) for a given claim and prompting our original baselines with these passages instead of the full book text.

As seen in Table 22, these RAG baselines (denoted by BM25 + model name) outperform our original LLaMA and ProLong baselines (+9%), but not the Qwen baseline (-15%). Compared to our CLIPPER models, however, these RAG baselines still lag by 22-50%. It is important to note that RAG approaches do not consistently outperform long-context models in long-form claim verification. For instance, Karpinska et al. (2024) and Kim et al. (2024) benchmark a similar RAG setup where GPT-4o is provided only with BM25-retrieved passages. As shown in Table 3 of Karpinska et al. (2024), the RAG versions (k = 5, 25, 50) consistently underperform the setting where GPT-4o has no retrieval support.

## D.6 Performance margin of CLIPPER

Table 23 lists statistical test results for the performance reported in Table 2. For CLIPPER-test, NoCha, and MuSR, which return binary True/False predictions, we use McNemar's test McNemar (1947). For NarrativeQA and InfiniBenchQA, which return ordinal scores ranging from 0 to 3, we use the Wilcoxon signed-rank test Wilcoxon (1992).

Fine-tuning on CLIPPER yields statistically significant improvements across all models on CLIPPER-test and NoCha. For MuSR, both Qwen and LLaMA show significant gains,

| Models | Groundedness |
|---|---|
| Qwen-Instruct | 11.9% |
| Llama-Instruct | 16.8% |
| ProLong-Instruct | 19.6% |
| Qwen-CLIPPER | 67.1% |
| Llama-CLIPPER | 75.9% |
| ProLong-CLIPPER | **80.6%** |

Table 21: Percentage of grounded chain of thoughts being generated by baseline and fine-tuned models. Our fine-tuned models generate much more grounded chain of thoughts.

| Model | CLIPPER-test (%) |
|---|---|
| Llama-3.1-8B-Instruct | 27.9 |
| ProLong-512K-8B-Instruct | 34.5 |
| Qwen2.5-7B-Instruct | 51.0 |
| BM25 + Llama-3.1-8B-Instruct | 36.45 |
| BM25 + ProLong-512K-8B-Instruct | 40.0 |
| BM25 + Qwen2.5-7B-Instruct | 36.0 |
| Llama-CLIPPER | 76.0 |
| ProLong-CLIPPER | 75.0 |
| Qwen-CLIPPER | 73.9 |

Table 22: Performance on CLIPPER-test for various models.

while ProLong does not. For ∞BenchQA, Qwen demonstrates a statistically significant improvement. For NarrativeQA, no models exhibit a significant improvement.

While improvements on NarrativeQA, MuSR, and ∞BenchQA are modest, these results represent performance on OOD tasks in our paper. NarrativeQA and ∞BenchQA focus on question answering over narrative contexts, while MuSR consists of algorithmically generated reasoning problems. Therefore, significant performance gains on these tasks would be nice to have, not expected.

| Baseline Models | CLIPPER-test ($\chi^2$) | NoCha ($\chi^2$) | NarrativeQA (Wilcoxon) | MuSR ($\chi^2$) | InfiniBenchQA (Wilcoxon) |
|---|---|---|---|---|---|
| Qwen2.5-7B-Instruct | 174.0 | 65.0 | 105.0 | 38.0 | 2825.5 |
| Llama-3.1-8B-Instruct | 0.0 | 52.0 | 205.5 | 36.0 | 2029.5 |
| ProLong-512K-8B-Instruct | 82.0 | 54.0 | 156.0 | 90.0 | 4556.5 |

Table 23: Test statistics comparing fine-tuned and baseline models across benchmarks. For CLIPPER-test, NoCha, and MuSR we report McNemar's $\chi^2$ statistic; for NarrativeQA and InfiniBenchQA we report the Wilcoxon signed-rank statistic. Table 24 shows p-value corresponding to these test statistics.

| Baseline Models | CLIPPER-test | NoCha | NarrativeQA | MuSR | InfiniBenchQA |
|---|---|---|---|---|---|
| Qwen2.5-7B-Instruct | 8.719614e−62 | 7.929521e−10 | 0.349212 | 0.000351 | 0.002420 |
| Llama-3.1-8B-Instruct | 6.406666e−145 | 1.467807e−12 | 0.791416 | 0.003866 | 0.544319 |
| ProLong-512K-8B-Instruct | 1.212688e−165 | 6.404008e−05 | 0.351747 | 0.316073 | 0.108244 |

Table 24: p-values for statistical significance (threshold $p < 0.05$). See Table 23 for test statistics.

