# OpenReview forum: "CLIPPER: Compression enables long-context synthetic data generation"
_colmweb.org/COLM/2025/Conference — COLM 2025_

### Official Review · Reviewer_hqHf · 2025-05-12

**Rating:** 8
**Confidence:** 4
**Ethics Flag:** 1

**Summary:**

The paper proposes CLIPPER, a new approach to generate synthetic data for long-context narrative claim verification. It involves a two-stage process: first, it compresses long documents into chapter outlines of a book, and then it generates claims based on these compressed representations. The claims can cover two expanses of data - chapter or the entire book. Additionally, they use LLM-as-a-judge to validate the correctness of the claims and the corresponding CoT. Continual pretraining on synthetic data generated by CLIPPER leads to significant improvements across different benchmarks.

**Reasons To Accept:**

1. Continual pretraining on synthetic data generated by CLIPPER leads to significant improvements across different benchmarks. Using such data is cheaper than getting equivalent human annotations, and enables scaling up at a cheaper cost.
2. Generating and then using CoT to train models means adding a veil of explainability to the models, which is desirable behaviour.

**Reasons To Reject:**

1. LLM-as-a-judge is used to check whether generated CoTs are grounded or not. While effective for cost and scale, it is unclear how good the judge would be at this task. On a small subset of data, authors need to correlate model judgments with human judgments, before asserting that such models can be effectively used as judges.

---

> ### Author Response · Authors · 2025-05-30
> **Authors' response to reviewer hqHf**
>
> We thank the reviewer for acknowledging the strength and interpretability of our pipeline.
>
> > LLM-as-a-judge is used to check whether generated CoTs are grounded or not. While effective for cost and scale, it is unclear how good the judge would be at this task. On a small subset of data, authors need to correlate model judgments with human judgments, before asserting that such models can be effectively used as judges.
>
> We would like to point the reviewer to **section A.6** and **Table 11** in the appendix, where we report how each LLM judge aligns with human annotations on 66 CoT samples. The LLM judge that we eventually choose, DeepSeek‑Distill‑Llama‑70B, has the highest agreement (>90%) with human annotators and provides clear explanations.

---

> > ### Comment · Reviewer_hqHf · 2025-06-10
> > **Small sample for LLM-judge and human correlations**
> >
> > Thanks for your response, authors! The sample size on which you perform this correlation analysis is too small to prove statistical significance. I would recommend that you do a larger study which can underline the reliability of your LLM judge system

---

### Official Review · Reviewer_yM7W · 2025-05-13

**Rating:** 7
**Confidence:** 4
**Ethics Flag:** 1

**Summary:**

The paper proposes a two-stage framework called CLIPPER for generating high-quality synthetic data for narrative claim verification. Unlike generating claims directly from the full raw text (which lead to tons of errors in the generated claims), CLIPPER first compresses books into chapter outlines and book summaries, and then generates claims based on those compressed text. Through systematic automatic and human evaluation, the authors show that the proposed method reduces error rates of generated claims significantly at lower cost. The authors then fine-tune different models on generated data with CLIPPER and achieve significant improvements on narrative claim verification and also other narrative understanding tasks (OOD).

**Questions To Authors:**

When constructing book-level claims, would those claims simply be conjunctions of claims for each chapter? How often would a claim naturally combine the subclaims from different chapters (instead of using words such as *and*)?

**Reasons To Accept:**

1. All claims in the paper are backed by thorough experiments and human evaluations

2. Experiments are designed well and the author also tests models’ generalizability on OOD tasks. Showing the usefulness of their synthetic data.

**Reasons To Reject:**

1. The way to generate reliable claims is basically leveraging hierarchical summarization. So the novelty is not that high, but the experiments and human evaluation is really thorough.

2. The authors should add performance margin such as in Table 2 to show whether the improvement is significant. It appears that the improvements on NarrativeQA and MuSR are minor.

3. The authors state that their fine-tuned models set a new state of the art for <10B models on long-context claim verification. The reviewer is curious about the performance of specialized claim verification models such as those trained for detecting errors in text summarization (breaking the long-context into chunks if the context length is limited for those models). Although those models are not trained on narrative claim verification, they also fall under the umbrella of checking the correctness of claims.

---

> ### Author Response · Authors · 2025-05-30
> **Authors' response to reviewer yM7W**
>
> We thank the reviewer for acknowledging the thoroughness of our experiments.
>
> > The way to generate reliable claims is basically leveraging hierarchical summarization. So the novelty is not that high, but the experiments and human evaluation is really thorough.
>
> Our approach goes beyond summarization to produce claims that are intentionally challenging for models to verify. In this sense, we adapt hierarchical summarization to the specific task of long-form claim verification, which we think is a valid and exciting way to demonstrate how established methods can be repurposed to solve challenging problems.
>
> > The authors should add performance margin such as in Table 2 to show whether the improvement is significant. It appears that the improvements on NarrativeQA and MuSR are minor.
>
> We present below statistical test results for the benchmarks reported in Table 2. For CLIPPER-test, NoCha, and MuSR, which return binary True/False predictions, we use McNemar’s test. For NarrativeQA and InfiniBenchQA, which return ordinal scores ranging from 0 to 3, we use the Wilcoxon signed-rank test.
>
> Fine-tuning on CLIPPER yields statistically significant improvements across all models on CLIPPER-test and NoCha. For MuSR, both Qwen and LLaMA show significant gains, while ProLong does not. For InfiniBenchQA, Qwen demonstrates a statistically significant improvement. For NarrativeQA, no models exhibit a significant improvement.
>
> We emphasize again that while improvements on NarrativeQA, MuSR, and InfiniBenchQA are modest, these results represent performance on OOD tasks in our paper. NarrativeQA and InfiniBenchQA focus on question answering over narrative contexts, while MuSR consists of algorithmically generated reasoning problems. Therefore, significant performance gains on these tasks would be nice to have, not expected.
>
> *Table 1. Test statistics comparing fine-tuned and baseline models across benchmarks*
> | Baseline Models          | CLIPPER-test (chi2) | NoCha (chi2) | NarrativeQA (wilcox stat) | MuSR (chi2) | InfiniBenchQA (wilcox stat) |
> |--------------------------|--------------|-------|-------------|------|---------------|
> | Qwen2.5-7B-Instruct      | 174.0        |  65.0 |       105.0 | 38.0 | 2825.5        |
> | Llama-3.1-8B-Instruct    | 0.0          | 52.0  | 205.5       | 36.0 | 2029.5        |
> | ProLong-512K-8B-Instruct | 82.0         | 54.0  | 156.0       | 90.0 | 4556.5        |
>
> *Table 2. p-values for statistical significance (p < 0.05)*
> | Baseline Models          | CLIPPER-test  | NoCha        | NarrativeQA | MuSR     | InfiniBenchQA |
> |--------------------------|---------------|--------------|-------------|----------|---------------|
> | Qwen2.5-7B-Instruct      | 8.719614e-62  | 7.929521e-10 |    0.349212 | 0.000351 | 0.002420      |
> | Llama-3.1-8B-Instruct    | 6.406666e-145 | 1.467807e-12 | 0.791416    | 0.003866 | 0.544319      |
> | ProLong-512K-8B-Instruct | 1.212688e-165 | 6.404008e-05 |    0.351747 | 0.316073 | 0.108244      |
>
> > The reviewer is curious about the performance of specialized claim verification models such as those trained for detecting errors in text summarization.
>
> We agree that specialized claim verification models are interesting baselines. However, our baselines, Qwen‑2.5‑7B‑Instruct, LLaMA‑3.1‑8B‑Instruct, and ProLong‑512K‑8B‑Instruct, are already among the strongest open‑weight models for claim verification, as shown in the NoCha benchmark. We believe that our baseline provides a meaningful upper bound for our task.
>
> We also provide results for OpenAI o1‑mini and Gemini 1.5‑Flash 8B, which are proprietary models of the comparable size as our CLIPPER models. These models outperform LLaMA and ProLong baseline but still lag behind Qwen‑2.5‑7B by around 3% and the worst CLIPPER model by 25% on our test set.
> |              Models             | CLIPPER‑test |  NoCha |
> |-------------------------------|------------|------|
> | OpenAI o1‑Mini                   | 48.7 %       | 28.6 % |
> | Gemini 1.5 Flash 8B               | 45.0 %       | 26.2 % |
> | Llama‑3.1‑8B‑Instruct           | 27.9 %        | 16.5 % |
> | ProLong‑512K‑8B‑Instruct        | 34.5 %        | 16.9 % |
> | Qwen2.5‑7B‑Instruct             | 51.0%        | 24.1%  |
> | Llama‑3.1‑8B‑CLIPPER            | 76.0 %       | 32.2 % |
> | ProLong‑512K‑8B‑CLIPPER         | 75.0 %       | 32.3 % |
> | Qwen‑2.5‑7B‑CLIPPER             | 73.9 %       | 32.4 % |
>
> > When constructing book-level claims, would those claims simply be conjunctions of claims for each chapter?
>
> We perform a quick heuristic check on 20K claims by looking for coordinating conjunctions (e.g., "and," "but," "or," "nor," "so," "for," "yet") in the claim texts. We find that 77.12% of true claims and 74.19% of false claims are conjunctions of multiple chapter events.
>
> While this is somewhat expected, we note that the number of naturally occurring combinations is still significant, with up to approximately 5K such claims for each type in the dataset.

---

> > ### Comment · Area_Chair_Fx8J · 2025-06-09
> >
> > Hi Reviewer yM7W,
> >
> > Can you confirm that you've read the authors' rebuttal?
> >
> > Thanks, AC

---

> > > ### Comment · Reviewer_yM7W · 2025-06-10
> > >
> > > The reviewer have read the authors' response and would like to see those being added to the next version of the paper. The reviewer will keep the score and want to emphasize that the experiments in the paper are really solid.

---

### Official Review · Reviewer_jD3k · 2025-05-13

**Rating:** 8
**Confidence:** 4
**Ethics Flag:** 1

**Summary:**

This paper tackles synthetic data generation for long-context reasoning tasks. The key motivation is that existing methods relies on prompting LLMs over long document, which suffers from global reasoning. The proposed method instead first compresses an entire book into chapter outlines, then generate claim and chain-of-thought reasoning over the shorter outlines. The claims can be synthesized on both book-level that requires reasoning over at least two chapters, or on chapter-level which focuses on key events of one chapter. Human validation shows that the claims error rate as well as dollar cost are significantly lower than existing approach. Further experiments by fine-tuning open models on the synthesized dataset for reasoning tasks shows the effectiveness of using compressed data.

**Questions To Authors:**

- Have you considered using different LLMs to generate claims to promote diversity / avoid model specific biases?

**Reasons To Accept:**

- This work proposed a novel approach that shows strong results from both human validation as well as supervised fine-tuning experiments. The dataset could also benefit research in chain-of-thought reasoning.
- The dataset is first validated by human, then further proved to be useful by supervised fine-tuning. The experiment is thorough, especially the authors verified there is no concern on data-leakage, as project Gutenberg is a popular dataset and might have been seen in some of the LLMs.
- The paper is well-motivated and nicely written.

**Reasons To Reject:**

- The compression stage could largely affect the quality of the output dataset. This part is difficult to tune and subject to model biases and potentially hallucination. It is also difficult to verify because of the volume.

---

> ### Author Response · Authors · 2025-05-30
> **Authors' response to reviewer jD3k**
>
> We thank the reviewer for acknowledging the novelty of our approach and the thoroughness of our experiments.
>
> > The compression stage could largely affect the quality of the output dataset. This part is difficult to tune and subject to model biases and potentially hallucination. It is also difficult to verify because of the volume.
>
> The compression stage, where entire books are compressed into summaries and chapter outlines, could indeed be challenging to properly tune and validate. However, prior research has demonstrated that LLMs are capable of producing high-quality summaries of long documents [1][2].
>
> In addition, these compressed representations could still provide a strong foundation for claim generations, as most of CLIPPER claims are grounded in the original book (as determined by our human validation on a subset of 66 claims). Therefore, we consider this compression stage adequate for our purposes.
>
> > Have you considered using different LLMs to generate claims to promote diversity / avoid model specific biases?
>
> This is a good point. At the time of writing, Claude-3.5-Sonnet is the only model we find capable of generating diverse claims that remain grounded in the source texts. We expect that newer LLMs with stronger instruction-following capabilities could also be used to diversify claim generation, and we will leave this to future research to explore.
>
> [1] Chang, Yapei, Kyle Lo, Tanya Goyal, and Mohit Iyyer. "Booookscore: A systematic exploration of book-length summarization in the era of llms." https://openreview.net/forum?id=7Ttk3RzDeu
>
> [2] Kim, Yekyung, Yapei Chang, Marzena Karpinska, Aparna Garimella, Varun Manjunatha, Kyle Lo, Tanya Goyal, and Mohit Iyyer. "FABLES: Evaluating faithfulness and content selection in book-length summarization." https://openreview.net/forum?id=YfHxQSoaWU

---

### Official Review · Reviewer_WmVp · 2025-05-23

**Rating:** 6
**Confidence:** 3
**Ethics Flag:** 1

**Summary:**

The paper introduces CLIPPER, compression-based method that generates high-quality synthetic data for narrative claim verification by first summarizing books into outlines, then generating claims and reasoning chains. This approach significantly improves the quality and complexity of claims, achieving state-of-the-art performance on long-context reasoning tasks with sub-10B models.

Updates:

I have read the response and acknowledge that the RAG-based baseline has been implemented. However, incorporating the other discussion points would require substantial restructuring and that is strongly recommended. I also encourage the authors to be more precise in their claim-related wording. In light of the baseline inclusion, I am increasing my score by one point.

**Questions To Authors:**

Please address the weaknesses .

**Reasons To Accept:**

Strengths:
- Address the difficult problems of handling long context input for difficult tasks like claim verification
- Results show proposed approach consistently outperform baselines

**Reasons To Reject:**

Weaknesses:

- Lack of robust baselines: The baseline LLMs reported in Table 2 are neither fine-tuned nor instruction-tuned, making them inadequate for meaningful comparison. This limits the credibility of performance gains. A more rigorous baseline, such as a simplified version of CLIPPER based on passage retrieval and fine-tuning on passage-claim-response pairs, would help better isolate the contribution of the proposed method. Furthermore, comparisons with additional proprietary LLMs beyond Claud, such as GPT-4, Gemini, or Mixtral, could have provided a broader context for evaluating task difficulty.

- Unsubstantiated claims: Certain conclusions, such as the one in Section 4.3 ("Short-context claim data is less helpful"), lack robust empirical support. The claim is positioned as contradicting prior work (e.g., Dubey et al., 2024; Gao et al., 2024b), yet the evaluation setup does not convincingly demonstrate this, especially as the baselines referenced are not instruction-tuned or optimized for the task.

- Some sections suffer from a lack of logical flow, making them difficult to follow. For instance, Section 2.2 ("Naïve claim generation using book texts") appears abruptly and is followed immediately by the introduction of CLIPPER without sufficient contextual buildup. These sections are central to the paper’s contributions but are presented in a disjointed and poorly sequenced manner.

- The Related Work section references a large number of prior studies but does so in a cursory fashion, often without elaboration or critical analysis. This weakens the reader’s understanding of how the current work builds on or diverges from existing research.

---

> ### Author Response · Authors · 2025-05-30
> **Authors' response to reviewer WmVp**
>
> We thank the reviewer for their detailed review and for acknowledging the strength of our method.
>
> > A more rigorous baseline, such as a simplified version of CLIPPER based on passage retrieval and fine-tuning on passage-claim-response pairs, would help better isolate the contribution of the proposed method.
>
> This baseline sounds like a retrieval-augmented generation (RAG) pipeline, where a model retrieves relevant passages and verifies claims using only the retrieved evidence. We quickly implement one version of such pipeline, using BM25 to retrieve the top 50 relevant book passages (each no longer than 256 words) for a given claim and prompting our original baselines with these passages instead of the full book text.
>
> | **Model**                                | **CLIPPER-test** |
> |------------------------------------------|--------------|
> | Llama‑3.1‑8B‑Instruct                    | 27.9 %       |
> | ProLong‑512K‑8B‑Instruct                 | 34.5 %       |
> | Qwen2.5‑7B‑Instruct                      | 51.0%        |
> | BM25 + Llama‑3.1‑8B-Instruct            | 36.45%       |
> | BM25 + ProLong‑512K‑8B‑Instruct         | 40.0%        |
> | BM25 + Qwen2.5‑7B‑Instruct              | 36.0%          |
> | Llama‑3.1‑8B‑CLIPPER                    | **76.0 %**       |
> | ProLong‑512K‑8B‑CLIPPER                 | 75.0 %       |
> | Qwen‑2.5‑7B‑CLIPPER                     | 73.9 %       |
>
>
> As shown in the table above, these RAG baselines (denoted by BM25 + model name) outperform our original LLaMA and ProLong baselines (+9%), but not the Qwen baseline (-15%). Compared to our CLIPPER models, however, these RAG baselines still lag by 22-50%.
>
> It is important to note that RAG approaches do not consistently outperform long-context models in long-form claim verification. For instance, Karpinska et al., 2024 [1] and Kim et al., 2024 [2] benchmark a similar RAG setup where GPT-4o is provided only with BM25-retrieved passages. As shown in Table 3 of [1], the RAG versions (k = 5, 25, 50) consistently underperform the setting where GPT-4o has no retrieval support.
>
> > Certain conclusions, such as the one in Section 4.3 ("Short-context claim data is less helpful"), lack robust empirical support, especially as the baselines referenced are not instruction-tuned or optimized for the task.
>
> We would like to clarify that our claim specifically concerns the limitations of short-context claim verification data in the context of long-form claim verification. We choose our baselines for this specific task based on their performance on the NoCha benchmark ([https://novelchallenge.github.io/](https://novelchallenge.github.io/)). We are not trying to show that the claims made by Dubey et al., 2024, and Gao et al., 2024 are entirely incorrect; rather, we want to highlight a long-form task where their findings may not apply. While we maintain that our claim is valid, we will revise the section header to indicate that this limitation specifically concerns long-form claim verification.
>
> > Some sections suffer from a lack of logical flow, making them difficult to follow. For instance, Section 2.2 ("Naïve claim generation using book texts") appears abruptly and is followed immediately by the introduction of CLIPPER without sufficient contextual buildup.
>
> We present Section 2.2 as a straw-man proposal, which is a straightforward and intuitive approach but does not work well. We put this section before the introduction of CLIPPER to have a proper motivation for our method. That being said, we will revise the first paragraph of Section 2.2. And 2.3 so that there is a clearer flow.
>
> > The Related Work section references a large number of prior studies but does so in a cursory fashion, often without elaboration or critical analysis. This weakens the reader’s understanding of how the current work builds on or diverges from existing research.
>
> Despite space constraints, we did try our best to list all relevant work. Throughout the paper, we cite and engage with these works where appropriate. In a future revision, we will expand on the concluding sentence of each related work subsection to provide further clarification.
>
> [1] Karpinska, Marzena, Katherine Thai, Kyle Lo, Tanya Goyal, and Mohit Iyyer. "One thousand and one pairs: A" novel" challenge for long-context language models." https://aclanthology.org/2024.emnlp-main.948/
>
> [2] Kim, Yekyung, Yapei Chang, Marzena Karpinska, Aparna Garimella, Varun Manjunatha, Kyle Lo, Tanya Goyal, and Mohit Iyyer. "FABLES: Evaluating faithfulness and content selection in book-length summarization." https://openreview.net/forum?id=YfHxQSoaWU

---

> > ### Comment · Area_Chair_Fx8J · 2025-06-09
> >
> > Hi Reviewer WmVp,
> >
> > Can you confirm that you've read the authors' rebuttal?
> >
> > Thanks, AC

---

### Author Response · Authors · 2025-06-05
**Invitation to Engage in Discussion**

Dear Reviewers,

Thank you for your thoughtful and constructive feedback. We have considered your comments and made several updates to the paper. Below is a brief summary of our responses:

1. We have added additional baselines, including those using RAG assistance (BM25-retrieved passages) as well as proprietary models comparable in size to our fine-tuned models.
2. We have included statistical margins in the main results table.
3. We have revisited the paper’s organization and addressed concerns related to specific claims.

We invite you to review our responses and, if appropriate, acknowledge them or update your review. Your input will help both us and the chairs better understand the strengths and remaining areas for improvement in the paper.

Thank you,
Authors of Submission 860

---

### Decision · Program_Chairs · 2025-07-08

**Decision:**

Accept

**Comment:**

The paper proposes a synthetic data generation pipeline for narrative claim verification that can automatically produce complex claims. They address issues with LLM long-form discourse understanding by “compressing” (summarizing) fiction narratives into chapter outlines, then generating 19K claims from the chapter outlines. They demonstrate that models trained on this synthetic data significantly improve over baselines on existing narrative understanding benchmarks and at claim verification.

While the reviewers highlight the strong empirical results and use of human validation, there are concerns about the quality of compression and thoroughness of the automatic evaluation in the paper. However, most of the automatic evaluation concerns have been resolved by the authors’ rebuttal which incorporates strong proprietary model and RAG-based approaches. While the idea of discourse-aware summarization as an intermediate step is not new, the approach is clearly effective, appears easily reproducible and is of benefit to the research community. The authors have also promised to expand upon related work in their revised version. I would suggest the authors provide more analysis of the compression stage and possible errors (e.g. hallucination) with a small-scale human validation before publication. I recommend acceptance.